# OPTIMAL ACTION ABSTRACTION FOR IMPERFECT IN-FORMATION EXTENSIVE-FORM GAMES

## ABSTRACT

Action abstraction is critical for solving imperfect information extensive-form games (IIEFGs) with large action spaces. However, due to the large number of states and high computational complexity in IIEFGs, existing methods often focus on using a fixed abstraction, which can result in sub-optimal performance. To tackle this issue, we propose a novel Markov Decision Process (MDP) formulation for finding the optimal (and possibly state-dependent) action abstraction. Specifically, the state of the MDP is defined as the public information of the game, each action is a feature vector representing a particular action abstraction, and the reward is defined as the expected value difference between the selected action abstraction and a default fixed action abstraction. Based on this MDP, we build a game tree with the action abstraction selected by reinforcement learning (RL), and solve for the optimal strategy based on counterfactual regret minimization (CFR). This two-phase framework, named RL-CFR, effectively trades off computational complexity (due to CFR) and performance improvement (due to RL) for IIEFGs, and offers a novel RL-guided action abstraction selection in CFR. To demonstrate the effectiveness of RL-CFR, we apply the method to solve Heads-up No-limit (HUNL) Texas Hold'em, a popular representative benchmark for IIEFGs. Our results show that RL-CFR defeats ReBeL's replication, one of the best fixed action abstraction-based HUNL algorithms, and a strong HUNL agent Slumbot by significant win-rate margins $64 \pm 11$ and $84 \pm 17$ mbb/hand, respectively.

## 1 INTRODUCTION

The imperfect information extensive-form game (IIEFG) model (Streufert, 2021) is a general formulation for studying multi-player turn-taking games in a tree representation, including Poker (Moravčík et al., 2017), MahJong (Wang, 2023) and Scotland Yard (Schmid et al., 2021). Solving IIEFGs requires finding the Nash equilibrium (Nash, 2002) of the game, especially under two-person zero-sum conditions. In recent years, the most popular approach for solving large IIEFGs has been counterfactual regret minimization (CFR) or its variants (Burch & Bowling, 2013; Tammelin, 2014; Brown & Sandholm, 2019b; Brown et al., 2019), which gives a mixed strategy with low exploitability for IIEFG.

However, many IIEFGs have myriad actions. As a result, the size of the game tree increases exponentially with the number of actions (Schnizlein et al., 2009), and directly applying CFR-based solutions encounters tremendous computational complexity. To mitigate this problem, action abstraction (Aceto, 1991), which selects limited number of actions from all available actions, has thus been extensively applied to greatly reduce the size of the game tree so that CFR can be solved efficiently. Nevertheless, due to the large number of different states and high computational complexity in IIEFGs, existing results mostly focus on fixed action abstractions (Moravčík et al., 2017; Brown et al., 2020; Zarick et al., 2020). Doing so inevitably lead to sub-optimality, and it remains an unsolved challenge to design strategies that can find optimal dynamic action abstractions with manageable computational complexity (Brown, 2020). In this paper, we propose an action abstraction technique that achieves better performance compared to fixed action abstraction methods, and the size of our action abstraction not exceed the fixed action abstraction.

Reinforcement learning (Humphreys, 1997; Sutton & Barto, 1998) (RL) has been shown to be a revolutionary method in many games (Madeira et al., 2006), e.g., Go (Silver et al., 2016), StarCraft

II (Lee et al., 2018) and Dota 2 (Berner et al., 2019). However, applying RL methods to IIEFGs is challenging due to two important features of IIEFG: (i) the optimal strategy for an IIEFG is most likely a mixed strategy on its support (Chen & Ankenman, 2007; Neyman, 2008), and (ii) the value of an information set may depend on the strategy that it is chosen (Brown et al., 2020). To see this, consider the simplified poker example (Burch, 2017) in Figure 1. In this case, player 1 has equal chance of being dealt $J$ or $K$ and player 2 is always dealt $Q$. There are 2 chips in the pot (both player put 1 chips in the pot), and both players have 2 chips left behind, with player 1 acts first. The Nash equilibrium strategy for player 1 is all-in all of $K$ and $50\%$ of $J$, and checking the other $50\%$ of $J$. If player 1 declares all-in, then the Nash equilibrium strategy for player 2 is calling and folding with equal probabilities. If a Nash equilibrium strategy is adopted, player 1's $K$ expects to win 2 chips, while $J$ expects to lose 1 chips, and player 2 expects to lose $0.5$ chips. If player 1 all-in with $100\%$ probability, and player 2's best response strategy is call with $100\%$ probability, player 1's $K$ expects to win 3 chips, while $J$ expects to lose 3 chips, and player 2 expects to win 0 chips.

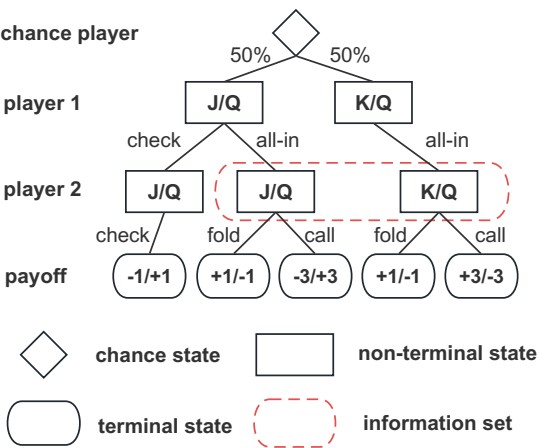

Figure 1: The game starts with a chance state where player 1 has equal chance of being dealt $J$ or $K$, and player 2 is always dealt $Q$. Player 1 will always all-in with $K$ and has to decide whether to check or all-in with $J$. If player 1 declares all-in, player 2 has to decide whether to fold or call. Player 2 does not know player 1's cards, the information set contains two states player 2 incapable of distinguishing. In the terminal state, we assign payoffs to both players based on the assignment rule.

To tackle the above challenges, we propose a two-phase framework, named RL-CFR, which ingeniously combines deep reinforcement learning (DRL) and CFR. Specifically, we first formulate a novel Markov Decision Process (MDP) (van Otterlo & Wiering, 2012) for determining the action-abstraction with highest expected payoff. For this MDP, the state is the *public information* of the game, each control action is a feature vector representing a particular action abstraction, and the action rewards are set to be the value differences calculated by CFR between selected action abstractions and default fixed action abstraction. Based on this MDP, we then build a game tree according to action abstraction selected by the actor-critic DRL method (Konda & Tsitsiklis, 1999), and eventually solve the strategy for selected action abstraction based on CFR. Our RL-CFR framework offers a principled way to reaps benefits from both RL and CFR, and handles the aforementioned mixed-strategy and probability-dependent reward issues. It also effectively trades off computational complexity (due to CFR) and performance improvement (due to RL) for IIEFGs.[1] As we will see in the experiments, *RL-CFR can be trained from scratch* given only the rules of the IIEFG. Compared to other methods for choosing action abstractions (Hawkin et al., 2011; 2012; Zarick et al., 2020), RL-CFR has a wider range of applicability and faster convergence (Brown, 2020).

To demonstrate the effectiveness of RL-CFR on large IIEFGs, we evaluate its performance on the challenging Heads-up No-limit Texas Hold'em (HUNL) poker game.[2] Our results show that RL-CFR defeats the fixed action abstraction-based HUNL algorithm ReBeL's replication (Brown et al., 2020) by $64$ mbb/hand win-rate in a test of over $600,000$ hands, and beats the popular strong HUNL agent Slumbot (Jackson, 2013) by $84$ mbb/hand win-rate in a test of over $250,000$ hands. These significant win-rate margins clearly show the power of our novel RL-CFR solution.

---

[1]The "trade off" is that action abstraction techniques reduce CFR's computational complexity of large IIEFGs, while RL-CFR achieves performance improvement through selecting an action abstraction that has a higher expected value calculated by CFR compared to the fixed action abstraction, and the size of this selected action abstraction does not exceed the size of the fixed action abstraction.

[2]Heads-up No-limit Texas Hold'em is a two-player form of Texas Hold'em, and is an important version of Texas Hold'em for investigating mixed strategy two-player zero-sum IIEFGs (Bard et al., 2013) due to its complex nature (Rubin & Watson, 2011) and extremely large decision space (Johanson, 2013).

The main contributions of our work are summarized as follows.

- We introduce a novel MDP formulation for IIEFGs, whose states are carefully defined based on public information, actions are feature vectors representing action abstractions, and rewards are value differences between selected action abstractions and default fixed action abstractions. The MDP formulation allows us to dynamically adjust the action abstraction at different states.

- Based on our novel MDP, we propose a novel framework RL-CFR, which effectively combines DRL with CFR to achieve a good balance between computation and optimism, and can be trained from scratch given only the rules of the IIEFG. RL-CFR effectively handles the large decision space and computational complexity of IIEFGs, and enables one to tradeoff computational complexity (due to CFR) and performance improvement (due to RL).

- We evaluate RL-CFR on the popular HUNL game. Our results show that RL-CFR defeats ReBeL's replication (one of the best fixed action abstraction-based HUNL algorithms) and Slumbot (the strongest publicly available HUNL AI provides online comparisons) by significant win-rates, i.e., by margins $64 \pm 11$ and $84 \pm 17$ mbb/hand, respectively.

## 2    RELATED WORK ON EXTENSIVE-FORM GAMES

**Methods of solving IIEFGs.** CFR-based algorithms (Burch & Bowling, 2013; Tammelin, 2014; Brown & Sandholm, 2019b; Brown et al., 2019) are are commonly used to solve large IIEFGs, because the regret of CFR is bounded linearly with the game size (a more detailed description of CFR is presented in Appendix D). There are methods such as Hedge (Cesa-Bianchi & Lugosi, 2006) or excessive gap technique (Hoda et al., 2010) that theoretically converge faster than CFR.

**Faster convergence and better efficiency for solving large IIEFGs.** (Habara et al., 2023) combined excessive gap technique with CFR for accelerating the solving of large IIEFGs. (Liu et al., 2023) investigated RL regularization techniques in solving IIEFGs and proposed a regularization-based payoff function. (Meng et al., 2023) proposed an efficient deep reinforcement learning method to solve the problem of inaccurate state value estimation in large IIEFGs.

**Action abstraction in IIEFGs.** The action abstraction technique is able to quickly compute a strategy for an IIEFG and obtain a solution with theoretical bounds (Kroer & Sandholm, 2014; 2018; 2015). In IIEFGs with myriad actions, action abstraction can affect strategy quality in surprising ways (Waugh et al., 2009; Chen & Ankenman, 2007). A parametric method (Hawkin et al., 2011) has been proposed to find the optimal action abstraction in IIEFGs, and an iterative algorithm (Hawkin et al., 2012) has been introduced to adjust the action abstraction during iteration. However, these methods change the action abstraction of each node in the game tree at each iteration, and therefore converge slower compared to fixed action abstraction methods (Brown, 2020; Zarick et al., 2020).

**RL for IIEFGs.** There are several CFR-based works inspired by RL, such as regression counterfactual regret minimization (Waugh et al., 2015; D'Orazio et al., 2020), neural fictitious self-play (Heinrich & Silver, 2016), deep counterfactual regret minimization (Brown et al., 2019) and ReBeL (Brown et al., 2020). (Pérolat et al., 2021) applied a regularization-based reward adaptation technique to solve two-player zero-sum IIEFGs with strong convergence guarantees. (Sokota et al., 2023) studied a RL algorithm called magnetic mirror descent to achieve empirically competitive results with CFR in two-player zero-sum games. (Pérolat et al., 2022) solved an imperfect information game Stratego with model-free multi-agent RL.

## 3    BACKGROUND AND NOTATION

**Imperfect Information Extensive-Form Games** We first provide the necessary notations for Imperfect Information Extensive-Form Games (IIEFGs) based on notations from (Streufert, 2021; Osborne & Rubinstein, 1994; Burch, 2017; Brown, 2020; Kovařík & Lis, 2019). Specifically, an IIEFG describes an imperfect information games in the form of a tree, and can be represented by $G = \langle \mathcal{H}, \mathcal{Z}, \mathcal{A}, \mathcal{N}, \mathcal{P}, \sigma_c, u, \mathcal{I} \rangle$, where each notation is explain below.

- $\mathcal{H}$ is the set of states (histories/nodes). A state $h \in \mathcal{H}$ is described by all history actions from the initial game state $\emptyset$. We use $\cdot$ to indicate concatenation, and $h \cdot a$ means taking an action $a$ at state $h$. $h \sqsubseteq h'$ means $h$ is an ancestor of $h'$, and $h \sqsubset h'$ means $h$ is a strict ancestor of $h'$.

- $\mathcal{Z} \subset H$ is the set of terminal states. A terminal state $z \in \mathcal{Z}$ has no available action.
- $\mathcal{A}(h) := \{a | h \cdot a \in \mathcal{H}\}$ is the set of available actions at a non-terminal state $h \in \mathcal{H} \backslash \mathcal{Z}$. $\mathcal{AA}(h) \subseteq \mathcal{A}(h)$ is an action abstraction for $\mathcal{A}(h)$.
- $\mathcal{N} = \{1, \cdots, N\}$ is the set of players. There is a "player" not in player set $\mathcal{N}$, defined as $c$, called chance decisions, which represents random events players can not control.
- A function $\mathcal{P} : \mathcal{H} \backslash \mathcal{Z} \to \mathcal{N} \cup \{c\}$ determines the acting player at a non-terminal state $h$. $\mathcal{H}_p$ is the set of all states $h$ such that $\mathcal{P}(h) = p$, and $\mathcal{H}_c$ is the set of chance states.
- The chance strategy $\sigma_c(h, a)$ is a probability that chance will act $a \in \mathcal{A}(h)$ at a state $h \in \mathcal{H}_c$.
- $u = (u_p)_{p \in \mathcal{N}}$ is the value function for each terminal state $z$.
- The information-partition $\mathcal{I} = (\mathcal{I}_p)_{p \in \mathcal{N}}$ describes the imperfect information of $G$ where $\mathcal{I}_p$ is a partition of $\mathcal{H}_p$ for each player $p$. A set $I \in \mathcal{I}_p$ is called an information set, and all states in $I$ are indistinguishable for player $p$. We denote $I(h)$ as the unique information set that contains $h$. There is a constraint that $\forall I \in \mathcal{I}_p, \forall h \in I$, we have same acting player $p$, same available actions $\mathcal{A}(h) := \mathcal{A}(I(h))$ and same action abstraction $\mathcal{AA}(h) := \mathcal{AA}(I(h))$.

A behaviour strategy $\sigma_p \in \Sigma_p$ is a function $\sigma_p(I, a) \in \mathbb{R}$ that determines a probability distribution over available actions $a \in \mathcal{A}(I)$ for every information set $I \in \mathcal{I}_p$. We denote $\sigma(I, a) = \sigma_{\mathcal{P}(I)}(I, a)$. $\sigma = (\sigma_p)_{p \in \mathcal{N}}$ is a strategy profile. $\pi^\sigma(h)$ is the probability of reaching state $h$ if players follow $\sigma$, calculated as $\pi^\sigma(h) = \prod_{h' \cdot a \sqsubseteq h} \sigma(h', a)$. $\pi_p^\sigma(h)$ is the probability of reaching state $h$ if players except $p$ take actions to $h$ and player $p$ follows $\sigma$. $\pi_{-p}^\sigma(h)$ is the probability of reaching state $h$ if player $p$ takes actions to reach $h$ and other players follow $\sigma$. The counterfactual value (CFV) (Zinkevich et al., 2007) for player $p$ of state $h$ is $v_p^\sigma(h) = \sum_{z \in \mathcal{Z}, h \sqsubseteq z} \pi_{-p}^\sigma(h) \pi^\sigma(z|h) u_p(z)$. The CFV of an information set $I \in \mathcal{I}_p$ is $v_p^\sigma(I) = \sum_{h \in I} (\pi_{-p}^\sigma(h) \sum_{z \in \mathcal{Z}, h \sqsubseteq z} (\pi^\sigma(z|h) u_p(z)))$.

**Public Belief State** Intuitively, the common knowledge of an IIEFG should include the player's strategies (Brown et al., 2020). Public belief state (PBS) is an assumption that treats players' strategies as common knowledge for reducing the state of large IIEFGs significantly, e.g., (Burch et al., 2014; Sustr et al., 2019; Kovařík & Lisý, 2019; Brown et al., 2020). Specifically, we define player $p$'s observation-action history (infostate) as $O_p = (I_1, a_1, I_2, a_2, \cdots)$,[3] which includes the information sets visited and actions taken by $p$. The unique infostate corresponding to a state $h \in \mathcal{H}_p$ for player $p$ is $O_p(h)$. The set of states that correspond to $O_p$ is denoted $\mathcal{H}(O_p)$. We use $\sim$ to denote states indistinguishable by some player, i.e., $g \sim h$ means $\bigvee_{i=1}^N O_i(g) = O_i(h)$ ($\bigvee$ is the OR operation on all expressions). A public partition is any partition $\mathcal{PS}$ of $\mathcal{H} \backslash \mathcal{Z}$ whose elements are closed under $\sim$ and form a tree Johanson et al. (2011). An element $PS \in \mathcal{PS}$ is called a public state that includes the public information that each player knows. The unique public state of a state $h$ and an infostate $O_p$ are denoted by $PS(h)$ and $PS(O_p)$, respectively. The set of states that match the public information of $PS$ is denoted as $\mathcal{H}(PS)$.

In general, a PBS $\beta$ is described by the joint probability distribution of the possible infostates of the players (Nayyar et al., 2013; Oliehoek, 2013; Dibangoye et al., 2016). Formally, given a public state $PS$, $\mathcal{O}_p(PS)$ is the set of infostates that player $p$ may be in, and $\triangle \mathcal{O}_p(PS)$ is a probability distribution over the elements of $\mathcal{O}_p(PS)$. Then, PBS $\beta = (\triangle \mathcal{O}_1(PS), \cdots, \triangle \mathcal{O}_N(PS))$.[4] The public state of PBS $\beta$ is denoted as $PS(\beta)$. The acting player at PBS $\beta$ is denoted $\mathcal{P}(\beta)$. The available actions for acting player at PBS $\beta$ is denoted $\mathcal{A}(\beta)$, and the action abstraction at PBS $\beta$ is denoted $\mathcal{AA}(\beta)$.

A subgame can be rooted at a PBS because PBS is a state of the perfect-information belief-representation game with well-defined values (Brown et al., 2020). At the beginning of a subgame, a history is sampled from the probability distribution of the PBS, and then the game plays as if it is the original game. The value for player $p$ of PBS $\beta$ (PBS value) when all players play according to $\sigma$ is $v_p^\sigma(\beta) = \sum_{h \in \mathcal{H}(PS(\beta))} (\pi^\sigma(h|\beta) v_p^\sigma(h))$. The value for an infostate $O_p \in \beta$ when all players play according to $\sigma$ is $v_p^\sigma(O_p|\beta) = \sum_{h \in \mathcal{H}(O_p)} (\pi^\sigma(h|O_p, \beta_{-p}) v_p^\sigma(h))$ where $\pi^\sigma(h|O_p, \beta_{-p})$ is the prob-

---

[3]Observation-action history is a kind of information set introduced in (Burch et al., 2014).

[4]In general, PBS can shed extraneous history to refine information (Brown et al., 2020). For example, in HUNL, we do not need to record the entire history of actions, and a PBS state contains chips information, position information, public cards, and the probability of private hands of two players. We can represent a PBS with a marginal probability distribution in HUNL example as shown in the equation.

ability of reaching state $h$ according to $\sigma$ assuming $O_p$ is reached and the probability distribution over infostates for player other than $p$ is $\beta_{-p}$.

# 4 A NOVEL MDP FORMULATION FOR IIEFGS

Below, we present our novel MDP formulation for IIEFGs. It is important to note that our formulation is an abstract MDP model, designed to determine the action abstraction of IIEFGs, based on which we perform a CFR algorithm to solve for the optimal strategy. Thus, it does not correspond exactly to the actual game dynamics in IIEFGs.

In general, a Markov Decision Process (van Otterlo & Wiering, 2012) (MDP) consists of the tuple $\langle \mathbf{S}, \mathbf{A}, P, r, \gamma \rangle$, where $\mathbf{S}$ is the set of states, $\mathbf{A}$ is the set of actions, $r : \mathbf{S} \times \mathbf{A} \mapsto \mathbb{R}$ is the reward function, $P(\mathbf{s}'|\mathbf{s}, \mathbf{a})$ is the state transfer function, and $\gamma$ is the discount factor. The objective is to find an optimal control policy $\pi^*$, which determines $\mathbf{a}_t = \pi^*(\mathbf{s}_t)$ at each time, to maximize the expected cumulative reward $R = \mathbb{E}\{\sum_{t=0}^{\infty} \gamma^t r(\mathbf{s}_t, \mathbf{a}_t)\}$.

**New state, action and reward function for IIEFGs.** We now specify the state $\mathbf{s}$, the action $\mathbf{a}$ and the value $r$ of our MDP formulation. Our design is inspired by (Brown et al., 2019), which transforms high-dimensional public belief states into low-dimensional public states.

(**State**) The dimension of the PBS is generally large because it needs to record the distribution of infostates. To reduce the dimension of the state, we use the public state as the state in MDP for a PBS $\beta$, denoted as $\mathbf{s} = PS(\beta)$. The public state needs to record public information known to both players, and the dimensionality is generally not very large.[5] The selection of public states has the additional advantage that the public states of the non-root nodes are fixed during the CFR iterations, while the PBS of the non-root nodes can change during the CFR iterations.

(**Action**) We design novel control actions for our abstract MDP to represent different action abstractions in the IIEFGs, based on which we will build a game tree for the CFR solving.

In IIEFG, there are some actions that are common and are added to the action abstraction no matter what the PBS $\beta$ is. We define such a set of actions as always-selected action set $\mathcal{AA}_{always}(\beta)$, and $\mathcal{AA}_{always}(\beta)$ can consist of a few of the most common actions in the set of available actions $\mathcal{A}(\beta)$. $\mathcal{AA}_{always}(\beta)$. Meanwhile, we also define a *default* fixed action abstraction $\mathcal{AA}_{base}(\beta)$ at PBS $\beta$. $\mathcal{AA}_{base}(\beta) \subseteq \mathcal{A}(\beta)$ is a set of actions related only to PBS $\beta$, and we have $\mathcal{AA}_{always}(\beta) \subseteq \mathcal{AA}_{base}(\beta)$. In general, $\mathcal{AA}_{base}(\beta)$ will be pre-specified to a set of available actions related to the important information of PBS $\beta$.

$\mathcal{AA}_{always}(\beta)$ and $\mathcal{AA}_{base}(\beta)$ can be chosen arbitrarily in any IIEFG, although different choices can affect the win-rate and running time, as mentioned in (Moravčík et al., 2017). For example, in HUNL experiments, we can choose $\mathcal{AA}_{always}(\beta) = \{F, C, A\}$ and $\mathcal{AA}_{base}(\beta) = \{F, C, A, 0.5 \times pot, 1 \times pot, 2 \times pot\}$ (same setting as (Moravčík et al., 2017; Brown et al., 2020)), where $F, C, A$ refer to fold, check/call and all-in respectively and $\times pot$ means the fraction of the size of the pot being bet. (Moravčík et al., 2017) shows the win-rate decreases by 96 mbb/hand after 10,000 hands if we make $\mathcal{AA}_{base}(\beta) = \{F, C, A, 1 \times pot\}$, although the running time improves by 6 times.

Action $\mathbf{a}$ in our MDP is used to select an action abstraction $\mathcal{AA}_{MDP}(\beta, \mathbf{a})$ at PBS $\beta$, and we describe the specifics next. Formally at PBS $\beta$, the action abstraction chosen by $\mathbf{a}$ is

$$\mathcal{AA}_{MDP}(\beta, \mathbf{a}) = \mathcal{AA}_{always}(\beta) \cup \mathcal{AA}_{optional}(\beta, \mathbf{a}) \tag{1}$$

where optional action set $\mathcal{AA}_{optional}(\beta, \mathbf{a})$ is the set of actions generated from the PBS $\beta$ and chosen action vector $\mathbf{a}$. Since the size of the game tree increases exponentially with the number of available actions, we can choose to have $K$ actions to the optional action set $\mathcal{AA}_{optional}(\beta, \mathbf{a})$. The action we design is a $2K$-dimensional vector $\mathbf{a} = (x_1, y_1, \cdots, x_K, y_K)$, with each dimension having a value between $-1$ and $1$. Precisely, $\mathcal{AA}_{optional}(\beta, \mathbf{a}) = \bigcup_{i=1}^{K} f(x_i, y_i, \beta)$, where $f(x_i, y_i, \beta)$ is a function to generate an available action from all available actions except actions in $\mathcal{AA}_{always}$ (specially, if $f(x_i, y_i, \beta) = \emptyset$, there is no action abstraction in this dimension) according to variable $x_i, y_i$ and PBS $\beta$. Since the set of available actions of IIEFGs with myriad actions tends to be

---

[5]For example, in HUNL, a public state only includes the previous actions of the two players, the public cards, chips in the pot, remaining chips and acting player. On the other hand, a public belief state includes 1,326 different private hands for both two players, and requires 2,652 more dimensions than a public state.

continuous, to define the function $f(x_i, y_i, \beta)$, we can correspond the continuous parameters $x_i, y_i$ one by one to the set of available actions $\mathcal{A}(\beta)$ of PBS $\beta$.

Below, we use HUNL as a concrete example to describe how to choose $K$ and $f(x_i, y_i, \beta)$. For HUNL, we set $K = 3$, which means we can select up to 3 raising scales other than all-in. Based on human experience and inspired by prior studies (Hawkin et al., 2011; 2012), a reasonable range for a raising scale other than all-in is $[0, 5] \times$pot. Thus, we define the $f(x_i, y_i, \beta)$ function to be

$$f(x_i, y_i, \beta) = \begin{cases} CLIP(2.5(x_i + 1) \times pot), & y_i \geq 0; \\ \emptyset, & y_i < 0. \end{cases} \quad (2)$$

where $CLIP$ is a function that corresponds the nearest available raising scale.

(**Reward**) For our abstract MDP, the reward of each action needs to be obtained by solving two depth-limited subgames (Brown et al., 2018) (a technique for limiting the size of IIEFG, described in Appendix E) according to CFR-based algorithm ReBeL (Brown et al., 2020), as described in the Appendix F. We now describe how to compute the reward $r$ from the PBS $\beta$, the state vector $\mathbf{s}$ and the action vector $\mathbf{a}$.

We first build a game tree rooted at PBS $\beta$ with selected action abstraction $\mathcal{AA}_{MDP}(\beta, \mathbf{a})$ [6]. Based on ReBeL, we then obtain a strategy profile $\sigma_{MDP}$ for PBS $\beta$ that gives state transfers for all infostates corresponding to all non-leaf nodes of the subgame. Based on the calculated strategy profile $\sigma_{MDP}$, we can calculate PBS value $v_{\mathcal{P}(\beta)}^{\sigma_{MDP}}(\beta)$ for the acting player, which is the expected value calculated for the acting player on PBS $\beta$ (details of the PBS value calculation are in the last paragraph of Section 3). We then build another game tree rooted at PBS $\beta$ with default fixed action abstraction $\mathcal{AA}_{base}(\beta)$. Similarly, we obtain the strategy profile $\sigma_{base}$ for this game tree based on ReBeL, and compute the PBS value $v_{\mathcal{P}(\beta)}^{\sigma_{base}}(\beta)$ for the acting player.

Finally, we define the reward as the PBS value difference between the chosen action abstraction and the default action abstraction, denoted $r(\mathbf{s}, \mathbf{a}) = v_{\mathcal{P}(\beta)}^{\sigma_{MDP}}(\beta) - v_{\mathcal{P}(\beta)}^{\sigma_{base}}(\beta)$. The state transition of the MDP depends on the mixed strategy calculated by CFR, as described in Section 5.

## 5 RL-CFR FRAMEWORK

In this section, we present the RL-CFR framework, which builds upon the ReBeL algorithm (Brown et al., 2020), an efficient method for solving depth-limited subgame (Brown et al., 2018) mentioned in Appendix F.[7] In contrast to ReBeL, which selects a fixed action abstraction, RL-CFR selects a dynamic action abstraction via RL. As we will see in the experiments, doing so allows us to optimize over the set of action abstraction and achieve better performance. It is important to note that applying the DRL approach to IIEFGs is highly nontrivial. The key challenge comes from the fact that one has to decide a mixed strategy for all information sets (Burch, 2017; Brown, 2020), which is hard to calculate directly by RL approach.

RL-CFR framework is an end-to-end self-training reinforcement learning process, with the procedure shown in Figure 2. We now describe the sampling steps for RL-CFR framework: ① Starting from the initial PBS of the game, each time we handle a non-chance and non-terminal PBS $\beta$[8], and we compress a high-dimensional PBS $\beta$ into a low-dimensional public state $\mathbf{s}$ by the method in Section 4. ② Passing through the action network and add a Gaussian noise (for increasing exploration) to obtain action vector $\mathbf{a}$, and this action vector will be mapped to a specified action abstraction $\mathcal{AA}_{MDP}(\beta, \mathbf{a})$. ③ Building two depth-limited subgames rooted at $\beta$ according to the default action abstraction $\mathcal{AA}_{base}(\beta)$ and selected action abstraction $\mathcal{AA}_{MDP}(\beta, \mathbf{a})$ respectively. ④ Using ReBeL algorithm to solve the strategies and values of the two subgames. ⑤ Calculating PBS value difference as reward $r$, and adding RL data $\{\mathbf{s}, \mathbf{a}, r\}$ to the training data (denoted as $Data^{RL}$) for action and critic network. ⑥ Randomly choosing a subgame and following the corresponding strategy $\sigma(\beta)$ for state transition to a child PBS $\beta'$ next. Let $\beta = \beta'$, and repeat step ①. Algorithm 1 shows the formal procedure of sampling process.

---

[6]We use the selected action abstraction only for the root.

[7]ReBeL can be used to obtain efficient solutions for large IIEFGs by self-play RL and CFR.

[8]If we encounter a chance PBS where acting player is chance player, we let the chance player randomly act and update the PBS. If we encounter a terminal PBS, the epoch of sampling ends.

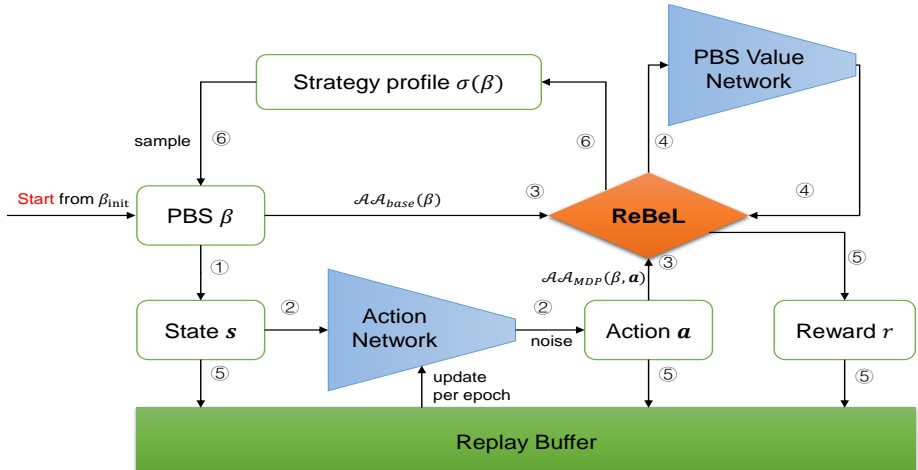

Figure 2: Training procedure for the RL-CFR framework. The labels in the figure correspond to the sampling steps for RL-CFR framework. A sampling epoch starts from the initial PBS $\beta_{init}$.

---

**Algorithm 1:** RL-CFR framework: Sampling $(s, a, r)$ data

**Input:** $\theta_\alpha, noise, \eta, \epsilon$ // $\eta = 0.33, \varepsilon = 0.25$ during training
$\beta \leftarrow \beta_{init}$
$Data^{RL} \leftarrow \{\}$
**while** *!IsTerminal*$(\beta)$ **do**
    **while** $P(\beta) = c$ **do**
        $\beta \leftarrow$ TakeChance$(\beta)$// Random chance event
    $\sigma_{base}(\beta), v_{\mathcal{P}(\beta)}^{\sigma_{base}}(\beta) \leftarrow$ReBeL$(\beta, \mathcal{AA}_{base}(\beta))$ // calculate the strategy and
        value for fixed default action abstraction
    $s \leftarrow PS(\beta)$ // use public state as the state in MDP
    $\mathbf{a} \leftarrow$ActionNetwork$(s, \theta_\alpha) + \mathcal{N}(0, noise)$ // $noise = 0.15$ during training
    $\sigma_{MDP}(\beta), v_{\mathcal{P}(\beta)}^{\sigma_{MDP}}(\beta) \leftarrow$ReBeL$(\beta, \mathcal{AA}_{MDP}(\beta, \mathbf{a}))$// calculate the strategy
        and value for selected action abstraction
    $r \leftarrow v_{\mathcal{P}(\beta)}^{\sigma_{MDP}}(\beta) - v_{\mathcal{P}(\beta)}^{\sigma_{base}}(\beta)$// reward function
    Add $\{\mathbf{s}, \mathbf{a}, r\}$ to $Data^{RL}$
    $c \sim unif[0, 1]$
    $d \sim unif[0, 1]$
    **if** $c < \eta$ **then**
        **if** $d < \epsilon$ **then**
            $nextaction \sim \mathcal{AA}_{base}(\beta)$
        **else**
            $nextaction \sim \sigma_{base}(\beta)$
        $\beta \leftarrow$NEXTPBS$(\beta, \sigma_{base}(\beta), nextaction)$
    **else**
        **if** $d < \epsilon$ **then**
            $nextaction \sim \mathcal{AA}_{limit}(\beta, \mathbf{a})$
        **else**
            $nextaction \sim \sigma_{MDP}(\beta)$
        $\beta \leftarrow$NEXTPBS$(\beta, \sigma_{MDP}(\beta), nextaction)$
**Output:** $Data^{RL}$

---

Thus, after each epoch, we sample a trajectory of $(\mathbf{s}_1, \mathbf{a}_1, r_1, \mathbf{s}_2, \mathbf{a}_2, r_2, \cdots, \mathbf{s}_t, \mathbf{a}_t, r_t)$ based on the current action network, where $t$ is the length of the game and depends on the player actions. After collecting a number of RL data $Data^{RL} = \{(\mathbf{s}, \mathbf{a}, r)\}$ in several epochs, we use the Actor-Critic algorithm (Konda & Tsitsiklis, 1999) and MSE Loss to train the action network and critic network

(These network structures are described in the Section 6). The loss function is as follows:

$$\mathcal{L}(\theta_c) = \mathbb{E}_{(\mathbf{s},\mathbf{a},r) \sim Data^{RL}}[(r^{\theta_c}(\mathbf{s},\mathbf{a}) - r)^2], \quad \mathcal{L}(\theta_a) = \mathbb{E}_{(\mathbf{s},\mathbf{a},r) \sim Data^{RL}}[-r^{\theta_c}(\mathbf{s}, \mathbf{a}^{\theta_a}(s))], \quad (3)$$

where $\theta_c, \theta_a$ are the parameters of critic network and action network.

In the early epochs of training, the action network selects an action abstraction that is often not as good as the default fixed action abstraction $\mathcal{AA}_{base}(\beta)$. Thus, we begin the training by choosing the default action abstraction when building the depth-limited subgame except for the root node. After a period of training the action abstraction chosen by the action network will outperform the default action abstraction, at which point, when building the depth-limited subgame, we choose the action abstraction for the child nodes based on the action network. Meanwhile, in order to get a more accurate PBS value, we can retrain the PBS value network according to the action abstraction selected by the action network. Theoretically, the PBS value network and action network can be repeatedly updated for training.

## 6 EXPERIMENT

To illustrate the effectiveness of our RL-CFR framework on large IIEFGs with myriad actions, we conduct experiments on Heads-up No-limit Texas Hold'em (HUNL), similar to many prior studies on large IIEFGs (Brown et al., 2019; 2020; Zarick et al., 2020). The rules of HUNL are provided in Appendix B. During the evaluation, both players start with 200 big blinds, and the two players will switch their positions and private hands in every two hands, as similarly done in annual computer poker competition (ACPC) (Bard et al., 2013).

Our experiments were run on a compute server with 4 NVIDIA PH402 SKU 200 GPUs and an 80-core Intel(R) Xeon(R) Gold 6145 2.00GHz CPU. All neural networks in our implementation consist of MLPs (size specified below) with ReLU (Glorot et al., 2011) activation functions, and are trained with Adam (Kingma & Ba, 2015). In the CFR iteration to solve a PBS, we use leading equilibrium-finding algorithm discounted CFR (DCFR) (Brown & Sandholm, 2019b), and we let the number of iterations $T = 250$ during training and evaluation.[9]

A PBS value network has 6 layers and 18 million parameters, of which the input layer has $2,678$ dimensions (corresponding to all possible private hands of the two players and the public state information). Each hidden layer has $1,536$ dimensions and the output layer has $2,652$ dimensions (corresponding to all possible private hands of the two players). During the training process, we sample $4.8 \times 10^7$ PBS data in total. We randomly sampled data from the last $1 \times 10^7$ PBS data and set a learning rate of $1 \times 10^{-5}$ and a batch size of $512$ during training. The training process and the data sampling process are performed simultaneously. Specifically, the data generation process is run in parallel in 60 threads, and the training process is run continuously on a single GPU. After the training of PBS value networks, we obtained a replication version of ReBeL as a baseline. In addition, the PBS value networks used for all our experiments (including the PBS value network used for RL-CFR) are trained based on the default action abstraction.[10]

The action network and the critic network both have 3 layers and $2 \times 10^4$ parameters, with hidden layers of 128 and 96 dimensions. The training process has $2 \times 10^6$ epochs, each sampling approximately 10 RL data.[11] We randomly sample data from all RL data, and set a learning rate of $1 \times 10^{-5}$ and a batch size of $1,024$ in training. After $5 \times 10^5$ epochs, we generated PBS Data by building the game tree exactly according to the action abstraction given by the action network. We generate data in parallel on 60 threads while training on a single GPU. The training cost of action network and critic network is approximately 40% of the training cost of PBS value network.

We evaluate the head-to-head performance of RL-CFR and ReBeL's replication under the common knowledge in HUNL. In detail, the common knowledge in HUNL is that the agent knows each

---

[9]Since HUNL evaluations are generally time-limited and need to be solved within a few seconds, common poker AIs typically take between 100 to 1000 CFR iterations (Brown et al., 2015; Bowling et al., 2017; Brown & Sandholm, 2017a; Moravcík et al., 2017; Brown et al., 2020).

[10]This setting is to illustrate that the performance improvement achieved by RL-CFR is entirely due to the action abstraction chosen by the action network.

[11]A RL Data $(\mathbf{s},\mathbf{a},r)$ consists of a 64-dimensional state $\mathbf{s}$ (record public cards, chips, positions and previous actions in HUNL), a 6-dimensional action $\mathbf{a}$ and a scalar $r$. Since the number of rounds in a HUNL game is not deterministic, a single sample to the terminate state will generally yield no more than 10 pieces of RL data.

other's hand ranges and previous actions played during the hand [12]. As shown in Table 1, after performing $600,000$ hands, RL-CFR achieves $64$ mbb/hand win-rate versus the replication of ReBeL.

We further compare RL-CFR against the open source AI Slumbot (Jackson, 2013), which was the winner of the 2018 ACPC and is the only HUNL AI we know that offers online competition testing. Since the opponent may select actions that deviate from the game tree, we perform nested subgame solving technique (Billings et al., 1998; Brown & Sandholm, 2017b; Moravčík et al., 2017) mentioned in Appendix E. We play RL-CFR for $250,000$ hands against Slumbot, and the test results are shown in Table 1, which illustrate that the replication of ReBeL beat Slumbot with a win-rate of 16 mbb/hand, while RL-CFR beat Slumbot with a win-rate of $84$ mbb/hand, and the win-rate of RL-CFR against Slumbot improved by $68$ mbb/hand relative to ReBeL. Note that a win-rate of over $50$ mbb/hand in poker is called a significant win-rate (Bowling et al., 2017), and RL-CFR clearly achieves significant win-rate against ReBeL and Slumbot.

Table 1: Competition results of the HUNL AIs against each other, measured in mbb/hand (variance was reduced by AIVAT technique (Burch et al., 2018)).

| AI name | ReBeL (replication) | Slumbot |
|---|---|---|
| ReBeL (replication) | - | $16 \pm 16$ |
| ReBeL (Brown et al., 2020) | - | $45 \pm 5$ |
| **RL-CFR** | $64 \pm 11$ | $84 \pm 17$ |

We also conducted an exploitability evaluation in over $10,000$ random river stage states.[13] The exploitability of a strategy $\sigma$ and a player $p$ is calculated by $expl_p(\sigma) = u_p^\sigma - \min_{\sigma^* \in \Sigma_{-p}} u_p^{\langle \sigma_p, \sigma^* \rangle}$ Cesa-Bianchi & Lugosi (2006). RL-CFR's exploitability is 17 mbb/hand and ReBeL's exploitability is 20 mbb/hand. The results indicate that RL-CFR generates action abstractions that are also less likely to be exploited in the context of generating more win-rate.

We perform additional experiments for RL-CFR, the results are shown in Table 2 Here we play against the method of choosing an optimal action abstraction among multiple fixed action abstractions (MUL-ACTION). MUL-ACTION works by choosing an action abstraction that has the greatest value to the root PBS $\beta_r$ among three action abstractions $\mathcal{AA}_{base1}(\beta_r), \mathcal{AA}_{base2}(\beta_r), \mathcal{AA}_{base3}(\beta_r)$[14]. We see that RL-CFR beats MUL-ACTION by $21 \pm 26$ mbb/hand win-rate after $100,000$ hands and only requires 1/3 of running time.

We also compare RL-CFR against choosing a finer-grained action abstraction (FINE-GRAIN) $\mathcal{AA}_{base}(\beta_r) = \{F, C, A, 0.33 \times pot, 0.5 \times pot, 0.75 \times pot, 1.0 \times pot, 1.25 \times pot, 2.0 \times pot\}$ at root PBS $\beta_r$ (the same setting as in (Zarick et al., 2020)). RL-CFR beats FINE-GRAIN by $23 \pm 28$ mbb/hand win-rate after $100,000$ hands and only requires approximately 4/7 of running time.[15]

Table 2: Win-rate and solving time of the method relative to RL-CFR.

| Method | Win-rate | Running time |
|---|---|---|
| ReBeL | $-64 \pm 11$ | $1\times$ |
| MUL-ACTION | $-21 \pm 26$ | $3\times$ |
| FINE-GRAIN | $-23 \pm 28$ | $1.75\times$ |

---

[12] HUNL can be modeled as a range-versus-range game (Kovarík & Lisý, 2019), and such common knowledge will not affect the evaluation (Burch et al., 2018) and will avoid nested subgame solving technique (Billings et al., 1998; Brown & Sandholm, 2017b; Moravčík et al., 2017).

[13] We simulate RL-CFR versus ReBeL until reaching river, i.e., the two agents choose their respective action abstractions, and the performance of the previously chosen action abstraction has no effect on the test results.

[14] We set $\mathcal{AA}_{base1}(\beta_r) = \{F, C, A, 0.5 \times pot, 1 \times pot, 2 \times pot\}, \mathcal{AA}_{base2}(\beta_r) = \{F, C, A, 0.25 \times pot, 0.5 \times pot, 1 \times pot\}, \mathcal{AA}_{base3}(\beta_r) = \{F, C, A, 0.33 \times pot, 0.7 \times pot, 1.5 \times pot\}$, and the action abstractions other than the root are the same as in ReBeL.

[15] Since the numbers of non-terminal nodes extended by the root node in the game tree built by FINE-GRAIN and RL-CFR are 7 and 4, respectively.

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

## A    CONCLUSIONS

In this work, we propose a novel algorithmic solution, named RL-CFR, for solving large-scale IIEFGs. RL-CFR builds on a novel abstract MDP formulation, which uses public information as states, action abstraction features as actions, and a carefully defined reward function. RL-CFR ingeniously combines reinforcement learning for action abstraction selection with CFR, to enable dynamic action abstraction selection in IIEFGs. Even though we cannot prove theoretically that RL-CFR has better convergence and less exploitable compared to the fixed action abstraction methods, through extensive experiments, we show that RL-CFR achieves a significant performance improvement compared to fixed action abstraction methods in HUNL benchmark.

## B    TEXAS HOLD'EM RULES

At the start of the game (hand), each player is given two cards, which we call "private hand". There are four stages in a game, called pre-flop, flop, turn and river, respectively. There are five public cards in a game, three cards are dealt[16] at the start of the flop and one card is dealt at the start of the turn and the river. Several players have to put in a pre-specified number of chips before the game starts, known as "small blind" and "big blind", and a "small blind" is usually half of a "big blind". In HUNL, the small blind player acts first in pre-flop stage, and the big blind player acts first in other stages. The legal actions are fold, check/call and bet/raise. In No-limit Texas Hold'em, players can bet/raise any number of chips between the last bet/raise in the stage (at least 1 big blind) and their remaining chips (all-in).

At the end of a game, each player who did not fold by the end of all stages chooses the best five cards out of two cards from private hand and the five public cards to compare, and the player (or several players) who has the best hand wins the pot. The win-rate in Texas Hold'em can be expressed as the average number of big blinds won per game, or in more granular units of mbb (Bowling et al., 2017) (one thousandth of a big blind). For example, we can say a win-rate of 0.01 big blind per hand, or 10 mbb/hand (10 mbb per hand).

## C    RELATED WORK OF TEXAS HOLD'EM AIS

The strongest Texas Hold'em AIs up to date, represented by Libratus (Brown & Sandholm, 2017a) and Pluribus (Brown & Sandholm, 2019a), have beaten top human players in both two-player and multi-player poker. They used complex abstraction (Johanson et al., 2012; Ganzfried & Sandholm, 2014; Brown et al., 2015) to reduce the huge decision space in Texas Hold'em and consume enormous computing resources to pre-calculate a blueprint strategy (Brown & Sandholm, 2016a) by CFR under a huge game tree.

---

[16]Poker term, means to open a new card.

DeepStack (Moravcík et al., 2017) employed deep learning to estimate values of private hands in game tree, thus reducing the size of the game tree and speeding up the CFR (Johanson, 2013). ReBeL (Brown et al., 2020) used self-play reinforcement learning (Heinrich, 2017) to generate more realistic training data than DeepStack. However, Libratus and Pluribus did not use reinforcement learning, and DeepStack and ReBeL did not use reinforcement learning in action abstraction selection (raising scales) in HUNL.

(Zhao et al., 2022) designed a HUNL AI based on reinforcement learning, which allows for an excellent AI with very few computational resources. However, since it did not use the widely used CFR algorithm in HUNL at all, it was fairly exploitable and had no theoretical guarantees.

## D COUNTERFACTUAL REGRET MINIMIZATION

Counterfactual Regret Minimization (CFR) is an algorithm for large IIEFGs that minimizes regret in each information set independently (Zinkevich et al., 2007), and can find $\varepsilon$-Nash equilibrium in two-player zero-sum IIEFGs.

Let $\sigma^t$ be the strategy profile of iteration $t$. The instantaneous regret for taking an action $a$ at information set $I \in \mathcal{I}_p$ on iteration $t$ is $r^t(I,a) = v_p^{\sigma^t}(I,a) - v_p^{\sigma^t}(I)$. The counterfactual regret for taking an action $a$ at $I$ on iteration $T$ is $R^T(I,a) = \sum_{t=1}^{T} r^t(I,a)$. The counterfactual regret is used for regret matching (RM) (Hart & Mas-Colell, 1997), a no-regret learning algorithm for solving imperfect-information game.

For an information set $I$, on each iteration $t+1$, an action $a \in \mathcal{AA}(I)$ is selected according to probabilities $\sigma^{t+1}(I,a) = \frac{R_+^t(I,a)}{\sum_{a' \in \mathcal{AA}(I)} R_+^t(I,a')}$ where $R_+^t(I,a) = \max\{0, R^t(I,a)\}$. If $\sum_{a' \in \mathcal{AA}(I)} R_+^t(I,a') = 0$, then we can chose an arbitrary strategy. In general, the upper bound on the regret value of the CFR or its variants (Burch & Bowling, 2013; Brown & Sandholm, 2017a; 2019b) is $O(L\sqrt{|\mathcal{AA}(I)|}\sqrt{T})$, where $L$ is the range of payoffs, $|\mathcal{AA}(I)|$ is the size of action abstraction for information set $I$ and $T$ is the number of iterations (Cesa-Bianchi & Lugosi, 2006).

Discounted CFR (DCFR) (Brown & Sandholm, 2019b) is the leading equilibrium-finding algorithm for large IIEFGs (Brown, 2020). DCFR is an variant of CFR with parameters $\alpha, \beta, \gamma$ (DCFR$_{\alpha,\beta,\gamma}$), defined by multiplying accumulated positive regrets by $\frac{t^\alpha}{t^\alpha+1}$, negative regrets by $\frac{t^\beta}{t^\beta+1}$ and contributions to the average strategy $\overline{\sigma}$ by $(\frac{t}{t+1})^\gamma$ on each iteration $t$. Our experiment setting is $\alpha = \frac{3}{2}, \beta = \frac{1}{2}$ and $\gamma = 2$, denoted $DCFR_{\frac{3}{2},\frac{1}{2},2}$.

## E ABSTRACTION

The huge solution complexity of IIEFGs is reflected in 3 dimensions: the depth of the game $D$, the size of the information set $|I|$ and the number of available actions $|\mathcal{A}(I)|$. In fact, the original space complexity is $O(|\mathcal{A}(I)|^D \cdot |I|)$, which is over the order of $10^{160}$ for HUNL with stacks of 200 big blinds and $20,000$ chips (Johanson, 2013). The time complexity of CFR to solve an IIEFG is $O(T \cdot |\mathcal{A}(I)|^D \cdot |I|)$ where $T$ is the number of iterations.

To limit the depth of the game, we generally do not compute the strategy to the end of the game, but instead generate a depth-limited subgame (Brown et al., 2018) that extends only a limited number of states into the future. We estimate the strategy or expected value of leaf states, which are non-terminal states in the full game but terminal states in the depth-limited subgame. DeepStack (Moravcík et al., 2017) and ReBeL (Brown et al., 2020) employs deep learning to estimate counterfactual values of leaf states, thus avoiding solving until the end of the game. Another way to limit the depth is consuming enormous computing resources to pre-calculate a blueprint strategy (Brown & Sandholm, 2016a; 2017a), thus avoiding solving when the game is deep.

To limit the size of the information set, we can put similar states into the same bucket (state-space abstraction) (Johanson et al., 2012; 2013; Brown et al., 2015) or represent the states in a high-dimensional feature abstraction (Brown et al., 2019).[17] State-space abstractions need to be carefully

---

[17]These methods also reduce the number of nodes in the game tree by putting similar nodes into same bucket.

designed to the specific game, and in order to illustrate the generality of our method to general EFGs, our experiments do not use any state-space abstractions.

To limit the number of available actions, it is common to use action abstraction in IIEFGs (Brown & Sandholm, 2016b). Formally, $\mathcal{A}\mathcal{A}(I)$ is the set of available actions at information set $I$, and $\mathcal{A}\mathcal{A}(I) \subseteq \mathcal{A}(I)$ is an action abstraction for $\mathcal{A}(I)$. If the opponent chooses an off-tree action $a$ that is not in the action abstraction $\mathcal{A}\mathcal{A}(I)$, we can round off-tree action to a nearby in-abstraction action (Schnizlein et al., 2009; Ganzfried & Sandholm, 2013) or resolve the strategy based on new action abstraction $\mathcal{A}\mathcal{A}(I) \cup \{a\}$ (nested subgame solving (Ganzfried & Sandholm, 2015; Brown & Sandholm, 2017b; Brown et al., 2020)).

## F SOLVING THE STRATEGY AND PBS VALUE FOR LARGE EFGS

In this section, we introduce the training process of ReBeL algorithm (Brown et al., 2020), a self-play RL method for solving the strategy and PBS values for large IIEFGs. Meanwhile, we take HUNL as an example to describe the setting of specific parameters.

In each epoch, we start training from the initial state of the game, and the PBS corresponding to the initial state is denoted as $\beta_{init}$. During training, we will deal with a PBS $\beta_r$ and corresponding action abstraction $\mathcal{A}\mathcal{A}(\beta_r)$. We need to compute the PBS value $v(\beta_r)$ and sample to the a leaf PBS $\beta_z$. Algorithm 2 shows the details and we describe the training process next.

---

**Algorithm 2:** ReBeL (Brown et al., 2020) algorithm[18]: Solving the strategy and PBS value for PBS $\beta_r$ with action abstraction $\mathcal{A}\mathcal{A}(\beta_r)$

---

**Function** *ReBeL*$(\beta_r, \mathcal{A}\mathcal{A}(\beta_r))$**:**

$G \leftarrow$ ConstructSubgame$(\beta_r, \mathcal{A}\mathcal{A}(\beta_r))$// construct a subgame with $\beta_r$ as the root

$\overline{\sigma}, \sigma^0 \leftarrow$ UniformPolicy$(\beta_r, \mathcal{A}\mathcal{A}(\beta_r))$

$\mathbf{v}(\beta_r) \leftarrow \mathbf{0}$

$t_{sample} \sim unif\{1, T\}$// Sample next iteration

**for** $t = 1 \cdots T$ **do**

  $G \leftarrow$ LeafValueEstimate$(G, \sigma^{t-1}, \theta)$// $\theta$ is the parameters of PBS value network

  $\sigma^t \leftarrow$ UpdatePolicy$(G, \sigma^{t-1})$

  $\overline{\sigma} \leftarrow \frac{t-1}{t+1}\overline{\sigma} + \frac{2}{t+1}\sigma^t$// Update average strategy based on $DCFR_{\frac{3}{2}, \frac{1}{2}, 2}$

  $\mathbf{v}(\beta_r) \leftarrow \frac{t-1}{t+1}\mathbf{v}(\beta_r) + \frac{2}{t+1}\mathbf{v}^{\sigma^t}(\beta_r)$// Update PBS value for all infostates at $\beta_r$

  **if** $t = t_{sample}$ **then**

    $\beta_{next} \leftarrow$ SampleLeaf$(G, \sigma^t)$// Sample a leaf PBS

Add $\{\beta_r, \mathbf{v}(\beta_r)\}$ to $Data^{PBS}$// Add PBS data for training

$v^{\overline{\sigma}}_{\mathcal{P}(\beta_r)}(\beta_r) \leftarrow$ ComputeValue$(\mathbf{v}(\beta_r))$// Compute PBS value for acting player at $\beta_r$

**return** $\overline{\sigma}, v^{\overline{\sigma}}_{\mathcal{P}(\beta_r)}(\beta_r), \beta_{next}$;

---

At the beginning of the training, we build a depth-limited subgame rooted with $\beta_r$.[19] In the process of building the game tree, when we deal with a non-terminal and non-leaf node $\beta'$, we expand the child nodes downwards according to the action abstraction $\mathcal{A}\mathcal{A}(\beta')$.[20]

After building the game tree, this subgame is solved by running $T$ iterations of CFR algorithm, and estimating the value of leaf nodes by a learned value network $\hat{v}$ at each iteration based on their PBS. On each iteration $t$, we first use CFR to determine a strategy profile $\sigma^t$ in the subgame. Next, the

---

[19]In the HUNL experiments, we build the subgame up to the end of the two players' actions in a stage or the end of the chance player's action. This means that an epoch has up to 7 phases, i.e., start of pre-flop, end of pre-flop, start of flop, end of flop, start of turn, end of turn and start of river.

[20]In the HUNL experiments, in order to reduce the size of the game tree, for the non-terminal PBS $\beta$ other than the root and root's sons, we set $\mathcal{A}\mathcal{A}(\beta) = \{F, C, A, 0.8 \times pot\}$.

infostate value of a leaf node $z$ is set to $\hat{v}(O_p(z)|\beta_z^{\sigma^t})$, where $\beta_z^{\sigma^t}$ is the PBS at $z$ when players play according to $\sigma^t$. Since the estimates of the neural network lead to a non-zero-sum game, we adjust the infostate values at each PBS so that the game satisfies zero-sum property. Also, for some infostates that shall have the same value under the rules of the game, we average their value estimates. PBSs may change every iteration, so the leaf node values may change every iteration. Given $\sigma^t$ and leaf node values, each infostate in each node has a calculated PBS value,[21] so that we can update the regret and average strategy $\overline{\sigma}$ for CFR algorithm.

After $T$ iterations, we solved the average strategy $\overline{\sigma}$. Based on this strategy, we calculate the PBS values for all infostates $v_p^{\overline{\sigma}}(O_p|\beta_r)$ for root PBS $\beta_r$, and denote this vector of PBS values as $\mathbf{v}(\beta_r)$. We then add the PBS data $\{\beta_r, \mathbf{v}(\beta_r)\}$ to the training data (denoted $Data^{PBS}$) for $\hat{v}(\beta_r)$. Meanwhile, we calculate the PBS value $v_{\mathcal{P}(\beta_r)}^{\overline{\sigma}}$ of $\beta_r$ according to calculated value vector $\mathbf{v}(\beta_r)$.[22]

Next, we sample a leaf PBS $\beta_z$ according to $\sigma^t$ on a random iteration $t \sim unif\{1, T\}$ where $T$ is the number of iterations, and to ensure more exploration, we can sample random leaf PBS with probability $\varepsilon$, and modify some public information in sampled PBS for more exploration[23]. We repeat above processes until the game ends.

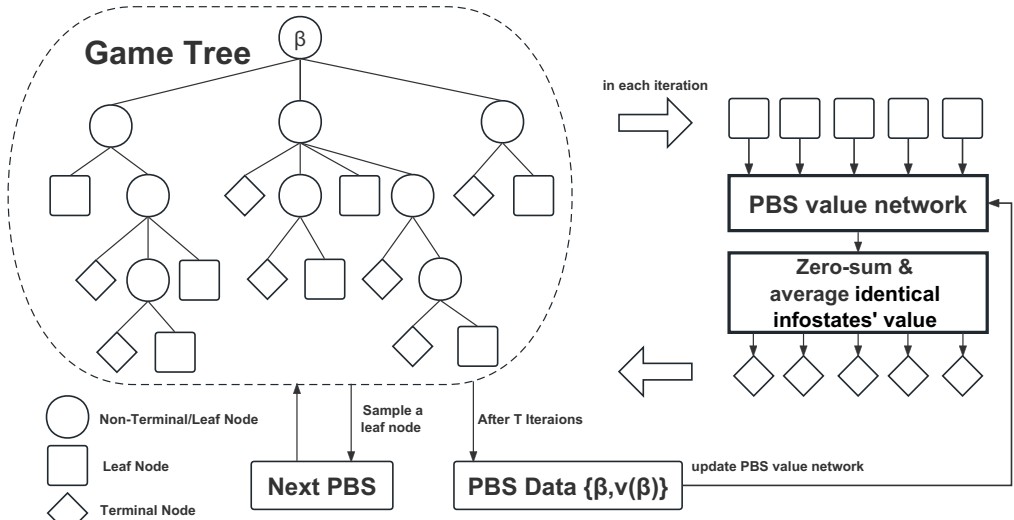

Figure 3: This figure shows how to generate PBS data and train PBS value network. For a PBS $\beta$, we build a depth-limit subgame rooted with $\beta$. A non-terminal and non-leaf node if represented by a circle, we expand the child nodes according to the action abstraction of the PBS of the node when we are building the game tree. A terminal node is represented by a diamond, we can directly calculate the PBS value for a terminal node. A leaf nodes is represented by a rectangle, and in each iteration of CFR we will use the PBS value network to estimate the PBS values of these leaf nodes (PBS values are re-estimated each iteration since they will be different each time we iterate to these nodes), then we can regard these leaf nodes as terminal nodes in this iteration.

We use Huber Loss (Huber, 1964) as the loss function for the PBS value network:

$$\mathcal{L}(\theta, \delta) = \mathbb{E}_{\{O_p, v_p(O_p)\} \sim \{\beta_r, \mathbf{v}(\beta_r)\}, \{\beta_r, \mathbf{v}(\beta_r)\} \sim Data^{PBS}}$$
$$[\min\{\frac{1}{2}(v_p(O_p) - \hat{v}^\theta(O_p|\beta_r))^2, \delta|v_p(O_p) - \hat{v}^\theta(O_p|\beta_r)| - \frac{1}{2}\delta^2\}] \tag{4}$$

where $\theta$ is the parameters of PBS value network, $O_p$ is an infostate in PBS $\beta_r$, and $\delta$ is a hyperparameter of Huber Loss.

---

[21]The details of calculating the PBS value are explained in Section 3.

[22]The ReBeL algorithm itself does not need to compute the PBS value $v_{\mathcal{P}(\beta_r)}^{\overline{\sigma}}$, but our RL-CFR framework requires this PBS value as part of the reward function.

[23]For HUNL agent training, we set $\varepsilon = 25\%$ for HUNL agent training, and for a sampled PBS, we multiply the chips in the pot by a random number between 0.9 and 1.1. For the PBS corresponding to the initial state, we set the chips of all players by a random number between 50 big blinds and 250 big blinds.

In summary, ReBeL is a self-play RL framework capable of continuously generating data from scratch for training, and Figure 3 shows the training process of ReBeL algorithm.

# G  EXAMPLES OF RL-CFR STRATEGIES

We use several examples from HUNL to illustrate how RL-CFR selects action abstractions. We show examples of heads-up evaluation between ReBeL's replication and RL-CFR. Both players start with 200 big blinds (BB) and $20,000$ chips (100 chips for 1 BB) in all examples.

**Example 1.**

*Pre-flop stage.* ReBeL sits in small blind position with hand 4♠3♠ and RL-CFR sits in big blind position with hand $J$♡8♢. ReBeL acts first with action abstraction $\{F, C, 2, 3, 5, A\}$[24]. The strategy calculated by CFR is: call with $3.21\%$, raise to 2 BB ($0.5\times$pot) with $52.10\%$, raise to 3 BB ($1\times$pot) with $44.11\%$ and raise to 4 BB ($2\times$pot) with $0.58\%$. ReBeL raises to 2 BB in this example. In this situation, RL-CFR selects an action abstraction $\{F, C, 3, 8.8, 16.21, A\}$. The strategy is: call with $76.08\%$, raise to 8.8 BB ($1.7\times$pot) with $23.67\%$ and raise to 16.21 BB ($3.5525\times$pot) with $0.25\%$. ReBeL calls in this example. When RL-CFR in the big blind is faced with a 2 BB raise, RL-CFR will use the three raising scales of 3 BB, 8.8 BB, 16.21 BB and expect to win 10 mbb/hand compared to the default raising scales (4 BB, 6 BB, 10 BB).

*Flop stage.* Flop is $J$♢6♡3♢. There are 4 BB in the pot and RL-CFR acts first. RL-CFR selects an action abstraction $\{F, C, A\}$. In this case, RL-CFR will check all hands, which is a common strategy that human professional players will employ. Now turn to ReBeL and the strategy is: check with $47.94\%$, bet 2 BB ($0.5\times$pot) with $51.63\%$ and bet 4 BB ($1\times$pot) with $0.42\%$. ReBeL bets 2 BB in this example. In this situation, RL-CFR selects an action abstraction $\{F, C, 4, 25.36, A\}$. The strategy is: call with $72.09\%$ and raise to 4 BB ($0.25\times$pot) with $27.91\%$. RL-CFR calls in this example. It's an interesting strategy, with RL-CFR opting for a minimum raising scale (mini-raise) and a very large raising scale, gaining an additional 6 mbb/hand win-rate compared to the default action abstraction.

*Turn stage.* Turn is 4♣. There are 8 BB in the pot and RL-CFR acts first. RL-CFR selects an action abstraction $\{F, C, 1, A\}$. The strategy is: check with $20.64\%$ and bet 1 BB with $79.36\%$. The turn card is very favourable to RL-CFR's calling range, so RL-CFR had a high frequency of betting (donk). Choosing a 1 BB raising scale (minimum betting) gives RL-CFR an additional win-rate of 42 mbb/hand compared to the default action abstraction, which is very impressive. RL-CFR bets 1 BB in this example. Now turn to ReBeL and the strategy is: call with $99.63\%$ and raise to 6 BB ($0.5\times$pot) with $0.37\%$. ReBeL has two-pairs now, however the strategy calculated by CFR is calling most hands since the turn card is unfavorable to small blind player's hand range.

*River stage.* River is 2♡. There are 10 BB in the pot and RL-CFR acts first. RL-CFR selects an action abstraction $\{F, C, 8.42, 14.88, 46.22, A\}$ with 4 mbb/hand extra win-rate. The strategy is: check with $99.93\%$ and bet 8.42 BB ($0.842\times$pot) with $0.07\%$. Now turn to ReBeL and the strategy is: check with $0.37\%$, bet 5 BB ($0.5\times$pot) with $43.76\%$, bet 10 BB ($1\times$pot) with $55.66\%$ and bet 20 BB ($2\times$pot) with $0.19\%$. ReBeL bets 5 BB in the example. Now turn to RL-CFR and the selected action abstraction is $\{F, C, 17.81, 55.31, 98.77, A\}$ with 6 mbb/hand extra win-rate. The strategy of RL-CFR is: fold with $42.04\%$, call with $56.52\%$, raise to 17.81 BB ($0.6405\times$pot) with $0.55\%$ and raise to 93.77 BB ($4.4385\times$pot) with $0.87\%$. RL-CFR folds and loses the 10 BB pot in the example.

**Example 2.** It is a symmetrical example of Example 1.

*Pre-flop stage.* RL-CFR sits in small blind position with hand 4♠3♠ and ReBeL sits in big blind position with hand $J$♡8♢. RL-CFR acts first and selects an action abstraction $\{F, C, 2.9, 4.56, 7.64, A\}$ with 24 mbb/hand extra win-rate. The strategy of RL-CFR is: call with $18.16\%$, raise to 2.9 BB ($0.95\times$pot) with $78.54\%$ and raise to 4.56 BB ($1.78\times$pot) with $3.29\%$. RL-CFR raises to 2.9 BB and ReBeL calls in this example.

*Flop stage.* Flop is $J$♢6♡3♢. There are 5.8 BB in the pot and ReBeL checks first. The action abstraction selected by RL-CFR is $\{F, C, 8.6, 28.1, A\}$. It is an interesting strategy generated by RL-CFR with overbet (bet more than a pot) only and gain an additional 46 mbb/hand win-rate

---

[24]$F, C, A$ refer to fold, check/call and all-in respectively and the numbers represent raising scales in BB.

compared to the default action abstraction. The strategy is: check with $99.65\%$, bet 8.6 BB with $0.22\%$ and bet 28.1 BB with $0.13\%$. RL-CFR checks in the example.

*Turn stage.* Turn is $4\clubsuit$. There are 5.8 BB in the pot and ReBeL acts first. The strategy of ReBeL is: check with $60.17\%$, bet 2.9 BB with $26.06\%$, bet 5.8 BB with $13.66\%$ and bet 11.6 BB with $0.12\%$. ReBeL checks in the example and turn to RL-CFR. The action abstraction calculated by RL-CFR is $\{F, C, 1, 1.85, A\}$. However, after evaluation by the policy network, RL-CFR considers this action abstraction to be inferior to the default action abstraction, so the default action abstraction will be chosen this time[25]. The strategy of RL-CFR is: check with $0.03\%$, bet 2.9 BB ($0.5\times$pot) with $98.88\%$, bet 5.8 BB ($1\times$pot) with $0.14\%$ and bet 11.6 BB ($2\times$pot) with $0.94\%$. In this example, RL-CFR bets 2.9 BB and ReBeL calls.

*River stage.* River is $2\heartsuit$. There are 11.6 BB in the pot and ReBeL checks first. The action abstraction selected by RL-CFR is $\{F, C, 2.3, 44.41, 46.43\}$ and the strategy of RL-CFR is: check with $0.30\%$, bet 2.3 BB with $99.64\%$ and bet $44.41, 44.63$ BB with $0.06\%$. In this example, RL-CFR bets 2.3 BB and ReBeL calls. RL-CFR wins the 16.2 BB pot with two pairs at showdown.

**Example 3.** In this example RL-CFR performed a bluff (betting with a weaker hand) and successfully bluffing with a suitable action abstraction.

*Pre-flop stage.* ReBeL sits in small blind position with hand $Q\heartsuit9\heartsuit$ and RL-CFR sits in big blind position with hand $9\clubsuit8\heartsuit$. ReBeL raises 2 BB first and RL-CFR calls in the example.

*Flop stage.* Flop is $K\clubsuit6\spadesuit2\diamondsuit$. There are 4 BB in the pot. RL-CFR and ReBeL check in the example.

*Turn stage.* Turn is $7\clubsuit$. There are 4 BB in the pot and RL-CFR acts first. The action abstraction selected by RL-CFR is $\{F, C, 1, 2.25, A\}$ with 10 mbb/hand extra win-rate. The strategy of RL-CFR is: check with $34.31\%$, bet 1 BB with $11.06\%$ and bet 2.25 BB with $54.63\%$. RL-CFR bets 1 BB in the example. The stragety of ReBeL is: fold with $48.32\%$, call with $51.04\%$, raises to 4 BB with $0.61\%$ and raises to 7 BB with $0.03\%$. ReBeL calls in the example.

*River stage.* River is $4\spadesuit$. There are 6 BB in the pot and RL-CFR acts first. The action abstraction selected by RL-CFR is $\{F, C, 4.56, 5.71, 29.23, A\}$ with 3 mbb/hand extra win-rate. The strategy of RL-CFR is: check with $12.48\%$, bet 4.56 BB with $41.63\%$, bet 5.71 BB with $33.11\%$ and bet 29.23 BB with $12.78\%$. RL-CFR bets 5.71 BB in the example and the strategy of ReBeL is: fold with $99.14\%$ and call with $0.85\%$. ReBeL folds and RL-CFR wins the 6 BB pot.

**Example 4.** In this example RL-CFR calls the 3-bet (re-raise at pre-flop) from ReBeL.

*Pre-flop stage.* RL-CFR sits in small blind position with hand $A\diamondsuit4\diamondsuit$ and ReBeL sits in big blind position with hand $K\spadesuit J\spadesuit$. RL-CFR raises to 2.9 BB. The strategy of ReBeL is: call with $2.37\%$, raises to 5.8 BB with $36.31\%$, raises to 8.7 BB with $42.01\%$ and raises to 14.5 BB with $19.31\%$. ReBeL raises to 8.7 BB in this example. The action abstraction selected by RL-CFR is $\{F, C, 14.5, 16.22, 27.96, A\}$ with 13 mbb/hand extra win-rate. The strategy of RL-CFR is: call with $70.90\%$, raise to 14.5 BB with $0.03\%$, raise to 16.22 BB with $0.04\%$, and raise to 27.96 BB with $29.03\%$. RL-CFR calls in the example.

*Flop stage.* Flop is $Q\clubsuit5\diamondsuit3\diamondsuit$. There are 17.4 BB in the pot and ReBeL acts first. The strategy of ReBeL is: check with $80.33\%$, bet 8.7 BB with $14.94\%$, bet 17.4 BB with $4.46\%$ and bet 34.8 BB with $0.26\%$. ReBeL bets 17.4 BB in the example. The action abstraction selected by RL-CFR is $\{F, C, 34.8, A\}$ with 142 mbb/hand extra win-rate, which means that if there is no mini-raise in the action abstraction there will be a huge loss in this situation. The strategy of RL-CFR is: call with $97.69\%$, raise to 34.8 BB with $1.73\%$ and all-in with $0.58\%$. RL-CFR calls in the example.

*Turn stage.* Turn is $6\diamondsuit$ and RL-CFR has the nuts (strongest hand). There are 52.2 BB in the pot and ReBeL checks first. The action abstraction selected by RL-CFR is $\{F, C, 15.11, 64.73, 97.22, A\}$ with 46 mbb/hand extra win-rate. The strategy of RL-CFR is: check with $0.43\%$, bet 15.11 BB with $88.13\%$ and bet 64.73 BB with $11.43\%$. RL-CFR bets 15.11 BB in the example and ReBeL folds. RL-CFR wins the 52.2 BB pot. The action abstraction of RL-CFR in this situation is very reasonable, and a 15.11 BB bet can put many of opponent's hands in an embarrassing situation.

---

[25]After selecting an action abstraction through the action network, we evaluate it using the policy network and if the evaluation value is negative, we use the default action abstraction.

