, where the state of the MDP is the *public information* of the game. For this MDP, each control action is a feature vector representing a particular action abstraction, and the action rewards are set to be the value differences calculated by CFR between selected action abstractions and default fixed action abstraction. Based on this MDP, we then build a game tree according to action abstraction selected by the actor-critic DRL method (Konda & Tsitsiklis, 1999), and eventually solve the strategy for selected action abstraction based on CFR. Our RL-CFR framework offers a principled way to reaps benefits from both RL and CFR, and handles the aforementioned mixed-strategy and probability-dependent reward issues. It also effectively trades off computational complexity (due to CFR) and performance improvement (due to RL) for IIEFGs. As we will see in the experiments, *RL-CFR can be trained from scratch* given only the rules of the IIEFG. Compared to other methods for choosing action abstractions (Hawkin et al., 2011; 2012; Zarick et al., 2020), RL-CFR has a wider range of applicability and faster convergence.

To demonstrate the effectiveness of RL-CFR on large IIEFGs, we evaluate its performance on the challenging Heads-up No-limit Texas Hold'em (HUNL) poker game.[1] Our results show that RL-CFR defeats the fixed action abstraction-based HUNL algorithm ReBeL (Brown et al., 2020) by 64 mbb/hand win-rate in a test of over $600,000$ hands, and beats the popular strong HUNL agent Slumbot (Jackson, 2013) by 84 mbb/hand win-rate in a test of over $250,000$ hands. These significant win-rate margins clearly show the power of our novel RL-CFR solution.

The main contributions of our work are summarized as follows.

- We introduce a novel MDP formulation for IIEFGs, whose states are carefully defined based on public information, actions are feature vectors representing action abstractions, and rewards are

---

[1]Heads-up No-limit Texas Hold'em is a two-player form of Texas Hold'em, and is an important version of Texas Hold'em for investigating mixed strategy two-player zero-sum IIEFGs (Bard et al., 2013) due to its complex nature (Rubin & Watson, 2011) and extremely large decision space (Johanson, 2013).

value differences between selected action abstractions and default fixed action abstractions. The MDP formulation allows us to dynamically adjust the action abstraction at different states.

- Based on our novel MDP, we propose a novel framework RL-CFR, which effectively combines DRL with CFR to achieve a good balance between computation and optimism, and can be trained from scratch given only the rules of the IIEFG. RL-CFR effectively handles the large decision space and computational complexity of IIEFGs, and enables one to tradeoff computational complexity (due to CFR) and performance improvement (due to RL).

- We evaluate RL-CFR on the popular HUNL game. Our results show that RL-CFR defeats ReBeL (one of the best fixed action abstraction-based HUNL algorithms) and Slumbot (the strongest publicly available HUNL AI provides online comparisons) by significantly win-rate margins, i.e., by margins $64 \pm 11$ and $84 \pm 17$ mbb/hand, respectively.

## 2 RELATED WORK ON EXTENSIVE-FORM GAMES

**Methods of solving IIEFGs.** CFR-based algorithms (Burch & Bowling, 2013; Tammelin, 2014; Brown & Sandholm, 2019b; Brown et al., 2019) are are commonly used to solve large IIEFGs, because the regret of CFR is bounded linearly with the game size (a more detailed description of CFR is presented in Appendix C). There are methods such as Hedge (Cesa-Bianchi & Lugosi, 2006) or excessive gap technique (EGT) (Hoda et al., 2010) that theoretically converge faster than CFR.

**Faster convergence and better efficiency for solving large IIEFGs.** (Habara et al., 2023) combines EGT with CFR for accelerating the solving of large IIEFGs. (Liu et al., 2023) investigates RL regularization techniques in solving IIEFGs and proposes a regularization-based payoff function. (Meng et al., 2023) proposes an efficient deep reinforcement learning method to solve the problem of inaccurate state value estimation in large IIEFGs.

**Action abstraction in IIEFGs.** In IIEFGs with myriad actions, one can gain more by choosing the right action abstraction (Chen & Ankenman, 2007). A parametric method (Hawkin et al., 2011) has been proposed to find the optimal action abstraction in IIEFGs, and an iterative algorithm (Hawkin et al., 2012) has been introduced to adjust the action abstraction during iteration. These methods of automatically selecting action abstraction for IIEFGs were also used in the early states of Libratus (Brown & Sandholm, 2017a) (one of the strongest HUNL AI). However, these methods change the action abstraction of each node in the game tree at each iteration, and therefore converge slower compared to fixed action abstraction methods (Brown, 2020; Zarick et al., 2020). As a result, these methods require a significant amount of time for pre-computation, and can generally only be used to a small number of early states in large IIEFGs.

## 3 BACKGROUND AND NOTATION

**Imperfect Information Extensive-Form Games** We first provide the necessary notations for Imperfect Information Extensive-Form Games (IIEFGs) based on notations from (Streufert, 2021; Osborne & Rubinstein, 1994; Burch, 2017; Brown, 2020; Kovaík & Lis, 2019). Specifically, an IIEFG describes an imperfect information games in the form of a tree, and can be represented by $G = \langle \mathcal{H}, \mathcal{Z}, \mathcal{A}, \mathcal{N}, \mathcal{P}, \sigma_c, u, \mathcal{I} \rangle$, where each notation is explain below.

- $\mathcal{H}$ is the set of states (histories/nodes). A state $h \in \mathcal{H}$ is described by all history actions from the initial game state $\emptyset$. We use $\cdot$ to indicate concatenation, and $h \cdot a$ means taking an action $a$ at state $h$. $h \sqsubseteq h'$ means $h$ is an ancestor of $h'$, and $h \sqsubset h'$ means $h$ is a strict ancestor of $h'$.
- $\mathcal{Z} \subset H$ is the set of terminal states. A terminal state $z \in \mathcal{Z}$ has no available action.
- $\mathcal{A}(h) := \{a | h \cdot a \in \mathcal{H}\}$ is the set of available actions at a non-terminal state $h \in \mathcal{H} \backslash \mathcal{Z}$. $\mathcal{A}\mathcal{A}(h) \subseteq \mathcal{A}(h)$ is an action abstraction for $\mathcal{A}(h)$.
- $\mathcal{N} = \{1, \cdots, N\}$ is the set of players. There is a "player" not in player set $\mathcal{N}$, defined as $c$, called chance decisions, which represents random events players can not control.
- A function $\mathcal{P} : \mathcal{H} \backslash \mathcal{Z} \to \mathcal{N} \cup \{c\}$ determines the acting player at a non-terminal state $h$. $\mathcal{H}_p$ is the set of all states $h$ such that $\mathcal{P}(h) = p$, and $\mathcal{H}_c$ is the set of chance states.
- The chance strategy $\sigma_c(h, a)$ is a probability that chance will act $a \in \mathcal{A}(h)$ at a state $h \in \mathcal{H}_c$.
- $u = (u_p)_{p \in \mathcal{N}}$ is the value function for each terminal state $z$.

- The information-partition $\mathcal{I} = (\mathcal{I}_p)_{p \in \mathcal{N}}$ describes the imperfect information of $G$ where $\mathcal{I}_p$ is a partition of $\mathcal{H}_p$ for each player $p$. A set $I \in \mathcal{I}_p$ is called an information set, and all states in $I$ are indistinguishable for player $p$. We denote $I(h)$ as the unique information set that contains $h$. There is a constraint that $\forall I \in \mathcal{I}_p, \forall h \in I$, we have same acting player $p$, same available actions $\mathcal{A}(h) := \mathcal{A}(I(h))$ and same action abstraction $\mathcal{A}\mathcal{A}(h) := \mathcal{A}\mathcal{A}(I(h))$.

A behaviour strategy $\sigma_p \in \Sigma_p$ is a function $\sigma_p(I, a) \in \mathbb{R}$ that determines a probability distribution over available actions $a \in \mathcal{A}(I)$ for every information set $I \in \mathcal{I}_p$. We denote $\sigma(I, a) = \sigma_{\mathcal{P}(I)}(I, a)$. $\sigma = (\sigma_p)_{p \in \mathcal{N}}$ is a strategy profile. $\sigma_{-p}$ is the strategy other than player $p$'s. $\pi^\sigma(h)$ is the probability of reaching state $h$ if players follow $\sigma$, calculated as $\pi^\sigma(h) = \prod_{h' \cdot a \sqsubseteq h} \sigma(h', a)$. $\pi_p^\sigma(h)$ is the probability of reaching state $h$ if players except $p$ take actions to $h$ and player $p$ follows $\sigma$. $\pi_{-p}^\sigma(h)$ is the probability of reaching state $h$ if player $p$ takes actions to reach $h$ and other players follow $\sigma$.

Under a strategy profile $\sigma$, player $p$'s expected value (EV) of a non-terminal state $h$ is $u_

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

---

**Function** $ReBeL(\beta_r, \mathcal{AA}(\beta_r))$**:**

$\quad G \leftarrow$ ConstructSubgame$(\beta_r, \mathcal{AA}limit(\beta_r))$

$\quad \overline{\sigma}, \sigma^0 \leftarrow$ UniformPolicy$(\beta_r, \mathcal{AA}(\beta_r))$

$\quad \mathbf{v}(\beta_r) \leftarrow \mathbf{0}$

$\quad t_{sample} \sim unif\{1, T\}$// `Sample next iteration`

$\quad$**for** $t = 1 \cdots T$ **do**

$\quad\quad G \leftarrow$ LeafValueEstimate$(G, \sigma^{t-1}, \theta)$// $\theta$ `is the parameters of PBS`

$\quad\quad\quad$ `value network`

$\quad\quad \sigma^t \leftarrow$ UpdatePolicy$(G, \sigma^{t-1})$

$\quad\quad \overline{\sigma} \leftarrow \frac{t-1}{t+1}\overline{\sigma} + \frac{2}{t+1}\sigma^t$// `Update average strategy based on` $DCFR_{\frac{3}{2}, \frac{1}{2}, 2}$

$\quad\quad \mathbf{v}(\beta_r) \leftarrow \frac{t-1}{t+1}\mathbf{v}(\beta_r) + \frac{2}{t+1}\mathbf{v}^{\sigma^t}(\beta_r)$// `Update PBS value for all`

$\quad\quad\quad$ `infostates at` $\beta_r$

$\quad\quad$**if** $t = t_{sample}$ **then**