# OpenReview forum: "Optimal Action Abstraction for Imperfect Information Extensive-Form Games"
_ICLR.cc/2024/Conference — Submitted to ICLR 2024_

### Official Review · Reviewer_Y7tW · 2023-10-26

**Soundness:** 3 good
**Presentation:** 3 good
**Contribution:** 3 good
**Rating:** 6
**Confidence:** 4

**Summary:**

The authors propose a new approach for performing action abstraction for solving large imperfect-information extensive-form games. In particular, instead of using a fixed action abstraction, they formulate the problem as a Markov decision process, and they employ techniques from reinforcement learning to gradually improve the action abstraction in conjunction with CFR. The overall algorithmic scheme is referred to as RL-CFR. The authors demonstrate that this new algorithm outperforms existing techniques for solving heads-up no-limit Poker.

**Strengths:**

Action abstraction is one of the most important modules when it comes to equilibrium computation in very large games. It has received considerable interest in the literature, but--as this paper demonstrates--there is room for improvement in the current techniques. Indeed, the authors propose an interesting and natural approach for performing dynamically action abstraction in a way that is compatible with modern RL techniques. They also demonstrate that their approach is promising through experiments on heads-up no-limit Poker, a standard benchmark in game solving with an enormous game-tree size.

**Weaknesses:**

In terms of the experimental evaluation, which is the key contribution of the paper, there are a couple of significant drawbacks. First, the paper focuses on a benchmark--namely heads-up no-limit Poker--that has been essentially already solved, meaning that prior work has already come up with techniques to find superhuman strategies in that game. It would be much more meaningful if the authors used their new algorithm to make progress on a benchmark that has been otherwise elusive using prior techniques. The second issue is that the comparison is not made with the state of the art models (Libratus and Deepstack). If I am not mistaken those models are not publicly available, but it is a significant weakness if the new approach does not attain state of the art performance. Do the authors know how Libratus or Deepstack performs against either Rebel or Slumbot? That would given an indication of how the new approach performs against Libratus or Deepstack. Remaining on the comparison drawn in the paper, one issue here is that Rebel was not tailored to Poker; the point of ReBel was to have an agent that performs well across many different benchmarks, so a comparison with an agent designed specifically for Poker might not be appropriate. Furthermore, Slumbot is not a particularly strong agent compared to either Libratus or Deepstack.

**Questions:**

Some minor points:

1. The way citations are used is not syntactically sound. For example, "(Meng et al.) proposes" should instead be "Meng et al. (2023) propose"
2. In page 8, "our conduct" is a typo
3. There is unnecessary space before Footnote 6
4. Section 6 gives too much detail that is not necessary in the main body; I would recommend delegating that information to the Appendix.

---

> ### Author Response · Authors · 2023-11-20
>
> Thank you very much for the valuable comments! We will answer your questions below.
> ***
> * Q1. First, the paper focuses on a benchmark--namely heads-up no-limit Poker--that has been essentially already solved, meaning that prior work has already come up with techniques to find superhuman strategies in that game. It would be much more meaningful if the authors used their new algorithm to make progress on a benchmark that has been otherwise elusive using prior techniques.
>
> * A1. As we mentioned in Footnote 2, HUNL is an important version of poker for investigating mixed strategy two-player zero-sum IIEFGs due to its complex nature and extremely large decision space. Although HUNL's AI has achieved superhuman performance, the problem of HUNL itself has still not been fully solved. We believe that all current HUNL AIs are still not achieving a Nash equilibrium strategy due to the extremely large decision space (more than $10^{160}$). Thus, we propose a general approach that can be used on IIEFGs and have made considerable progress on HUNL benchmark. On the other hand, we also agree with the reviewer that extending our method to more benchmarks is interesting and plan to study that in our future research.
> ***
> * Q2. The second issue is that the comparison is not made with the state of the art models (Libratus and Deepstack). If I am not mistaken those models are not publicly available, but it is a significant weakness if the new approach does not attain state of the art performance. Do the authors know how Libratus or Deepstack performs against either Rebel or Slumbot? That would given an indication of how the new approach performs against Libratus or Deepstack.
>
> * A2. We would like to first highlight that  the core contribution of our work is to propose a new method for general IIEFG to select good action abstractions with RL. Our novel approach has the potential to apply to problems that fall into the IIEFG setting. Here is for the comparison between our method and Libratus. BabyTartanian8 [1] is the champion of 2016 annual computer poker competition. Here are some head-up results against BabyTartanian8, ReBeL wins 9 mbb/hand, Slumbot loses 12 mbb/hand, Libratus wins 63 mbb/hand. Relative to BabyTartanian8, Libratus achieved about 54 mbb/hand win-rate more than ReBeL. For comparison, RL-CFR achieved a win rate of 64 mbb/hand in heads-up evaluation against the ReBeL's replication. This suggests that RL-CFR achieves a lift comparable to how Libratus compared to ReBeL.
> ***
> * Q3. Remaining on the comparison drawn in the paper, one issue here is that Rebel was not tailored to Poker; the point of ReBel was to have an agent that performs well across many different benchmarks, so a comparison with an agent designed specifically for Poker might not be appropriate. Furthermore, Slumbot is not a particularly strong agent compared to either Libratus or Deepstack.
>
> * A3. We would like to highlight that RL-CFR, like ReBeL, is not tailor-made for poker. Instead, it can be applied to choosing  appropriate action abstractions for general IIEFGs. As you mentioned, Slumbot is not a particularly strong agent compared to either Libratus or Deepstack. However, Slumbot is the only HUNL AI we know that offers online competition testing. Our comparison with Slumbot is mainly used to verify the reliability of our method. As shown in the experiments, RL-CFR achieves considerable win-rates against default action abstraction method ReBeL in both the heads-up evaluation and the indirect evaluation with the Slumbot.
> ***
> * Q4. Some minor points.
>
> * A4. Thank you very much for your help. We have fixed these typos.
> ***
>             [1] Noam Brown and Tuomas Sandholm. Baby tartanian8: Winning agent from the 2016 annual computer poker competition. In IJCAI, pages 4238–4239, 2016.
>
> We hope our responses address your concerns. Should you find the rebuttal satisfying, we wonder if you could kindly consider raising the score rating for our paper? We will also be happy to answer any further questions you may have. Thank you very much.

---

> > ### Comment · Reviewer_Y7tW · 2023-11-21
> > **Thank you for the Response**
> >
> > I thank the authors for the helpful response. I have increased my score accordingly.

---

> > > ### Author Response · Authors · 2023-11-21
> > >
> > > Thank you very much for raising the score rating! We really appreciate your prompt and positive feedback.

---

### Official Review · Reviewer_BWf4 · 2023-10-30

**Soundness:** 2 fair
**Presentation:** 2 fair
**Contribution:** 2 fair
**Rating:** 6
**Confidence:** 3

**Summary:**

This paper presents a system that uses actor-critic algorithms to learn an action abstraction used to reduce the size of imperfect-information extensive-form games to be solved with CFR.

The system is evaluated on Heads-up No-limit Texas Hold'em Poker, where it is compared with Rebel using a fixed action abstraction. Results show that the learned action abstraction allows for a final stronger strategy of Poker.

**Strengths:**

The idea of learning an action abstraction is interesting and valuable. It is also interesting that this is done by interplaying learning the action abstraction and solving the game with such an abstraction with a variant of CFR.

The quality of the action abstraction clearly plays a role in the quality of the strategy learned, so caring for the abstraction is a good research direction. If I understand it correctly, the key point in this work is to be able to solve the game with a small lookahead, to allow for a quick (but limited) evaluation of different action abstractions.

The empirical results are strong against a good baseline algorithm.

**Weaknesses:**

One weakness of the paper is that it uses much more notation than what is actually needed. The idea of connecting RL for learning action abstractions and CFR could be explained in a more pedagogical way. For example, it is not clear to me whether the definitions in the last paragraph of "Imperfect Information Extensive-Form Games" (Section 3) are needed at all. The pseudocode is provided, but the reader has to walk it through by themselves.

The experiments could be stronger if it considered a baseline that uses the abstraction by Hawkin et al. within the same Rebel framework. The paper hypothesizes that such a learning process would be slow because the abstraction changes in every iteration. To be honest, I don't see how this would be different with the proposed approach. Seeing learning curves of different versions of the system would dismiss any doubts about the contributions of the proposed method.

The paper makes it sound that this system can be used end-to-end with little to no human intervention. However, the method critically depends on human knowledge to even define the possible action abstractions, the set of "must-have" actions, and the default action abstraction that guides the RL process of learning a better abstraction. For example, how would the method behave if we removed the default action abstraction, so in the beginning of learning the method would do what RL algorithms do, which is to randomly select action abstractions?

**Questions:**

Instead of relying on RL to find the action abstractions, would it be possible to incorporate the search for action abstractions as part of the search Rebel does for an equilibrium? This would be equivalent to extending the depth of the tree, where a four player (in addition to player 1, 2, and the chance player) would attempt to maximize the value of the game. The decisions of this fourth player would be the selection of the abstraction used. How different would this be from the RL formulation in the current approach?

Instead of looking at the difference between the current action abstraction and the default abstraction, why not use a reward function that depends only on the value $v$ of the current abstraction?

---

> ### Author Response · Authors · 2023-11-20
>
> Thank you very much for the valuable comments! We will answer your questions below.
> ***
> * Q1. One weakness of the paper is that it uses much more notation than what is actually needed. The idea of connecting RL for learning action abstractions and CFR could be explained in a more pedagogical way. For example, it is not clear to me whether the definitions in the last paragraph of "Imperfect Information Extensive-Form Games" (Section 3) are needed at all. The pseudocode is provided, but the reader has to walk it through by themselves.
>
> * A1. Thank you for the suggestion. We have revised this paragraph, and only retained the counterfactual value which will be needed later.
> ***
> * Q2. The experiments could be stronger if it considered a baseline that uses the abstraction by Hawkin et al. within the same Rebel framework. The paper hypothesizes that such a learning process would be slow because the abstraction changes in every iteration. To be honest, I don't see how this would be different with the proposed approach. Seeing learning curves of different versions of the system would dismiss any doubts about the contributions of the proposed method.
>
> * A2. Faster convergence here means that compared to these previous methods of choosing dynamic action abstractions, RL-CFR has a faster convergence because it selects an action abstraction that is fixed during CFR solving. Our statement comes from Page 102 of [1], where it has been demonstrated that "the algorithm converges more slowly than running CFR with fixed bet sizes". We will consider using a baseline that uses the abstraction by Hawkin et al. within the same RebeL framework in future work. By the way, RL-CFR achieves a performance gain under the same solving time. To make this clear, we have made additional clarifications in the last sentence of Section 1, Paragraph 2 and Footnote 1 in the revision.
> ***
> * Q3. The paper makes it sound that this system can be used end-to-end with little to no human intervention. However, the method critically depends on human knowledge to even define the possible action abstractions, the set of "must-have" actions, and the default action abstraction that guides the RL process of learning a better abstraction. For example, how would the method behave if we removed the default action abstraction, so in the beginning of learning the method would do what RL algorithms do, which is to randomly select action abstractions?
>
> * A3. We believe that for an IIEFG with myriad actions, it is reasonable to set a simple default fixed action abstraction manually, and even if manually selected action abstractions are not used, it is possible to divide some of the action abstractions evenly according to the action space. For your question, if we removed the default action abstraction, it is difficult to evaluate the performance of an action abstraction, because the reward function is based on the default action abstraction.
> ***
> * Q4. Instead of relying on RL to find the action abstractions, would it be possible to incorporate the search for action abstractions as part of the search Rebel does for an equilibrium? This would be equivalent to extending the depth of the tree, where a four player (in addition to player 1, 2, and the chance player) would attempt to maximize the value of the game. The decisions of this fourth player would be the selection of the abstraction used. How different would this be from the RL formulation in the current approach?
>
> * A4. The approach you propose is very insightful. We used a similar approach to your proposition for MUL-ACTION in Table 2, trying to choose an action abstraction with the highest value out of multiple action abstractions. Currently, it seems that this method consumes much more time compared to the RL method, which is not desired, as the solving time is also a important metric for IIEFGs. We plan to design methods with multiple action abstractions without increasing the solving time in the future.
> ***
> * Q5. Instead of looking at the difference between the current action abstraction and the default abstraction, why not use a reward function that depends only on the value $v$ of the current abstraction?
>
> * A5. Because in an IIEFG, different states correspond to different except values. For example, in HUNL, a player in the small blind expects a positive payoff, while a player in the big blind expects a negative payoff. We are trying to design a value function that is not affected by the fact that these states themselves have different expected values.
> ***
>             [1] Noam Brown. Equilibrium Finding for Large Adversarial Imperfect-Information Games. PhD thesis, Carnegie Mellon University, 2020.
>
> We hope our responses address your concerns. Should you find the rebuttal satisfying, we wonder if you could kindly consider raising the score rating for our paper? We will also be happy to answer any further questions you may have. Thank you very much.

---

> > ### Author Response · Authors · 2023-11-21
> >
> > We have revised the paper according to reviewers' insightful comments and helpful suggestions, with revised parts marked in blue. To explain the reviewer's questions more intuitively, in Appendix G of the revision, we give several detailed examples of how RL-CFR selects an appropriate action abstraction.
> >
> > We wonder whether our responses address your concerns? Should you find the rebuttal satisfying, we wonder if you could kindly consider raising the score rating for our paper? We will also be happy to answer any further questions you may have. Thank you very much.

---

### Official Review · Reviewer_YRS4 · 2023-10-31

**Soundness:** 2 fair
**Presentation:** 1 poor
**Contribution:** 2 fair
**Rating:** 3
**Confidence:** 4

**Summary:**

The authors present a method called RL-CFR for dynamic action abstraction that uses reinforcement learning to find action abstractions as a function of the PBS. States in the MDP formulation are public belief states, and actions are potential action abstractions in those public belief states. A subgame with the chosen action abstraction is then approximately solved via ReBel, and rewards are based on the difference between the value of the PBS under the policy learned with the selected abstraction compared to its value from the policy learned via a base abstraction. Experiments are conducted in HUNL poker and show improved performance against strategies learned via ReBel with the base abstraction and Slumbot.

**Strengths:**

The MDP construction and application of RL to dynamic action abstraction is interesting and insightful. I believe the direction of the research is promising. Action abstraction is required to scale many of the algorithms for approximately solving imperfect information games with strong theoretical guarantees, so the work is well-motivated.

**Weaknesses:**

My main issues with the paper can be summarized as:
1. Failure to adequately summarize prior work in action abstraction. Some of the most important concepts (e.g. imperfect recall, and its effect on theoretical guarantees, is not discussed at all).
2. Lack of evidence on central claims in the paper.

For 1:
* **Waugh, Kevin, et al. "Abstraction pathologies in extensive games." AAMAS (2) 2009 (2009): 781-8.**
I believe the reader should be informed that abstraction can affect solution quality in surprising ways. While the authors mention that using a smaller $\mathcal{AA}_{base}$ as in Moravc´ık et al., 2017, leads to a decrease in win rate, it is also known that using a larger abstraction does not necessarily lead to improved solution quality. Waugh et al. discuss these abstraction pathologies in-depth here.

* **Kroer, Christian, and Tuomas Sandholm. "Extensive-form game abstraction with bounds." Proceedings of the fifteenth ACM conference on Economics and computation. 2014.** and **Kroer, Christian, and Tuomas Sandholm. "A unified framework for extensive-form game abstraction with bounds." Advances in Neural Information Processing Systems 31 (2018).** On the other hand, these papers discuss computing abstractions with bounds on their solution quality.

* ** Kroer, Christian, and Tuomas Sandholm. "Discretization of continuous action spaces in extensive-form games." Proceedings of the 2015 international conference on autonomous agents and multiagent systems. 2015.** The discretization of continuous action spaces may also be relevant to this line of work.

For 2:

The title of the paper including **Optimal** makes it seem like the paper will include theoretical guarantees on optimality of the learned abstraction, but it does not.

**"Compared to other methods for choosing action abstractions (Hawkin et al., 2011; 2012; Zarick et al., 2020), RL-CFR has a wider range of applicability and faster convergence."**
* I don't see evidence supporting either of these claims. The tested application in experiments is only poker, and no other abstraction algorithm is used.
* As an aside, the cost of running ReBel every time the action abstraction changes during RL seems enormous, and the result that the learned abstraction is better than a fixed one is not very surprising.

**"This two-phase framework, named RL-CFR, effectively trades off computational complexity (due to CFR) and performance improvement (due to RL) for IIEFGs"**
* Again, as far as I understand, there is evidence that RL-CFR increases performance at an increased cost, but I don't see evidence of the ability to trade off complexity for performance. The claim itself may be a bit unclear.

**Questions:**

The main question I have is regarding the cost of running RL-CFR. Is it possibly more efficient to run ReBel with a larger abstracted action set?

There were also some clarity issues in the paper.
* Public belief states and their importance could be explained better. I'm not sure what is meant by the following: "Public belief state (PBS) is an assumption(?) that treats players’ strategies as common knowledge for reducing the state of large IIEFGs significantly". I don't understand how a PBS reduces the size of the state space, and, as far as I understand, all variants of CFR treat strategies as common knowledge.
* In the last paragraph of section 3, it might help to explain *how* a PBS can be interpreted as a history in a perfect information "analogue" of some IIEFG.
* The claim "Which effectively combines DRL with CFR to achieve a good balance between computation and optimism" is difficult to interpret. I'm also not sure how it is supported.

General mistakes/typos I noticed:

* "so that CFR can still be solved efficiently"
* "A behaviour strategy"
* "our conduct experiments on"

---

> ### Author Response · Authors · 2023-11-20
>
> Thank you very much for the valuable comments! We will answer your questions below.
> ***
> * Q1. Failure to adequately summarize prior work in action abstraction. Some of the most important concepts (e.g. imperfect recall, and its effect on theoretical guarantees, is not discussed at all).
>
> * A1. Thank you for your comments, we have included these prior works to the background in revision.
> ***
> * Q2.  "Compared to other methods for choosing action abstractions (Hawkin et al., 2011; 2012; Zarick et al., 2020), RL-CFR has a wider range of applicability and faster convergence." I don't see evidence supporting either of these claims. The tested application in experiments is only poker, and no other abstraction algorithm is used.
>
> * A2. Faster convergence here means that compared to these previous methods of choosing dynamic action abstractions, RL-CFR converges faster  because it selects an action abstraction that is fixed during CFR solving. In page 102 of [1], it has been demonstrated "the algorithm converges more slowly than running CFR with fixed bet sizes". By the way, we have compared RL-CFR with some other abstraction algorithms listed in Table 2. A wider range of applicability means that RL-CFR constructs an MDP that solve a kind of IIEFGs, including but not limited to poker.
> ***
> * Q3. As an aside, the cost of running ReBel every time the action abstraction changes during RL seems enormous, and the result that the learned abstraction is better than a fixed one is not very surprising.
>
> * A3. We first train a PBS value network based on ReBeL, and then train networks for RL-CFR. The training cost of RL-CFR framework does not exceed the training cost of the PBS value network. At the evaluation time, RL-CFR is only used to select a fixed action abstraction for the current state, which requires only a one-time action network overhead and does not add additional computational cost to the CFR. ReBeL's default fixed action abstraction has 6 actions and RL-CFR selects an action abstraction that has up to 6 actions, and the action abstraction selected by RL-CFR will be fixed during CFR solving. Thus, RL-CFR's action abstraction will not increase the computational complexity compared to a default fixed action abstraction.
> ***
> * Q4. "This two-phase framework, named RL-CFR, effectively trades off computational complexity (due to CFR) and performance improvement (due to RL) for IIEFGs" Again, as far as I understand, there is evidence that RL-CFR increases performance at an increased cost, but I don't see evidence of the ability to trade off complexity for performance. The claim itself may be a bit unclear.
>
> * A4. RL-CFR has the same computational complexity compared to ReBeL (as stated in previous question), and achieves a performance gain under the same solving time. To make this clear, we have made additional clarifications in Footnote 1 in the revision.

---

> > ### Author Response · Authors · 2023-11-20
> >
> > ***
> > * Q5. The main question I have is regarding the cost of running RL-CFR. Is it possibly more efficient to run ReBel with a larger abstracted action set?
> >
> > * A5. As we mentioned in previous questions, the size of RL-CFR's action abstraction is no more than the size of default action abstraction in our experiments. A larger action abstraction would theoretically have a better strategy, but this would result in a larger game tree and more solving time, and the default action abstraction size we chose is relatively reasonable.
> > ***
> > * Q6. Public belief states and their importance could be explained better. I'm not sure what is meant by the following: "Public belief state (PBS) is an assumption(?) that treats players’ strategies as common knowledge for reducing the state of large IIEFGs significantly". I don't understand how a PBS reduces the size of the state space, and, as far as I understand, all variants of CFR treat strategies as common knowledge.
> >
> > * A6. In IIEFG, each state needs to include both the private information of the two players and the public information of the game. For the private information, because there are many different possibilities, a single enumeration can lead to the number of states becoming very large. In PBS, each state contains a group of states from the original IIEFG that are indistinguishable for at least one player.
> > ***
> > * Q7. In the last paragraph of section 3, it might help to explain how a PBS can be interpreted as a history in a perfect information "analogue" of some IIEFG.
> >
> > * A7. Based on your comments, we have made an additional clarification in revision about PBS in the Footnote 4.
> > ***
> > * Q8. The claim "Which effectively combines DRL with CFR to achieve a good balance between computation and optimism" is difficult to interpret. I'm also not sure how it is supported.
> >
> > * A8. We use RL to select a reasonable action abstraction and then use CFR to solve the strategy based on this action abstraction. Since the size of the action abstraction we choose does not exceed the size of the default action abstraction, RL-CFR can achieve performance improvement without increasing the solving time. This is also illustrated by the experimental results in Table 2.
> > ***
> >         [1] Noam Brown. Equilibrium Finding for Large Adversarial Imperfect-Information Games. PhD thesis, Carnegie Mellon University, 2020.
> >
> > We hope our responses address your concerns. Should you find the rebuttal satisfying, we wonder if you could kindly consider raising the score rating for our paper? We will also be happy to answer any further questions you may have. Thank you very much.

---

> > > ### Author Response · Authors · 2023-11-21
> > >
> > > We have revised the paper according to reviewers' insightful comments and helpful suggestions, with revised parts marked in blue. To explain the reviewer's questions more intuitively, in Appendix G of the revision, we give several detailed examples of how RL-CFR selects an appropriate action abstraction.
> > >
> > > We wonder whether our responses address your concerns? Should you find the rebuttal satisfying, we wonder if you could kindly consider raising the score rating for our paper? We will also be happy to answer any further questions you may have. Thank you very much.

---

> > > ### Comment · Reviewer_YRS4 · 2023-11-21
> > >
> > > Thank you for your response! Unfortunately, I don't see how the main concerns I've brought up have been addressed.
> > >
> > > Even though a citation for Waugh et al. (2009) has been included, its main point seems to contradict this paper's main contribution. Effectively, Waugh et al. are saying that larger abstractions do not necessarily lead to solutions that are closer to **optimal** in the real game. Yet this paper claims "This two-phase framework, named RL-CFR, **effectively trades off computational complexity (due to CFR) and performance improvement** (due to RL) for IIEFGs". These two ideas seem contradictory if "performance improvement" means closer to optimal and the tradeoff involves the number of actions in the abstraction.
> > >
> > > As Reviewer 3LiM pointed out, the paper does not discuss optimality in the traditional sense, which may be fine, but even in the latest revision of the paper, the language (e.g. in the title) may be misleading.
> > >
> > > On to the point about computational "complexity", which is also a confusing term here because it is used in a non-traditional sense to mean the training/evaluation cost of the learning algorithm, how can **RL-CFR have the same computational complexity compared to ReBeL**, yet be making a tradeoff between performance and complexity as stated above? I understand that evaluation time is not significantly increased, but training cost is the concerning factor anyway. How much computational effort is required to *find* the action abstraction that gets used at evaluation time?

---

> > > > ### Author Response · Authors · 2023-11-22
> > > >
> > > > Thank you for your prompt response! We appreciate your constructive comments.
> > > >
> > > > * First of all, we note that our claim does not controdict with that in Waugh et al. (2009). Instead, the strategy given by RL-CFR confirms the view of Waugh et al. "larger abstractions do not necessarily lead to solutions that are closer to optimal in the real game". This is shown in the examples in Appendix G, where we can obtain a better strategy by choosing a smaller action abstraction.
> > > >
> > > > * Second, as reviewers mentioned, we does not address any theory or optimality in the traditional sense. The main contribution of our work is to design a novel MDP formulation for IIEFGs, and propose an algorithmic framework based on RL and CFR to achieve good performance.
> > > >
> > > > * Third, the "trade off" in our statement does not mean that we achieve a better performance by increasing the number of actions in the action abstraction. Instead, we mean that RL-CFR can achieves performance improvement through better action abstraction, which could reduce CFR's "computational complexity" for large IIEFGs.   We agree with the reviewer that this should be made clearer to avoid confusion. Thus, we have clarified it in Footnote 1 of the new revision. As for the training cost, RL-CFR has an additional 40\% training cost compared to ReBeL's replication as shown in the experiment settings. We agree with the reviewer to clarify this, and we have stated this comparison explicitly in the last sentence in Section 6, Paragraph 4 of the new revision.
> > > >
> > > > We will also be happy to answer any further concerns you may have. Thank you very much.

---

### Official Review · Reviewer_3LiM · 2023-11-01

**Soundness:** 2 fair
**Presentation:** 2 fair
**Contribution:** 3 good
**Rating:** 6
**Confidence:** 4

**Summary:**

This paper proposes a reinforcement-learning-based method for picking action abstractions in imperfect-information extensive-form games (IIEFGs) like poker. (Action abstraction is mainly applied to the game of poker, where it means choosing the "discretization" of bet sizes into somewhere around 1 to 5 bet sizes.)

This paper extends the depth-limited search method ReBeL which uses self-play to train public belief state (PBS) value nets. In this paper, the novel algorithm RL-CFR takes in a pretrained PBS value net from running standard ReBeL self-play training. RL-CFR then does its own self-play training loop to train an action-abstracter RL agent. The self-play training loop is like the ReBeL self-play training loop, but the RL agent is being trained, not the PBS value function. On each move during the self-play training loop, the RL agent picks the action abstraction that will be used to do search for that move. Then ReBeL search is performed for the search tree. ReBeL search is also performed for the search tree that uses a default action abstraction. For each of the two solved search trees, we get some equilibrium value for the turn player. The reward to the RL agent is the difference between those two values.

The RL agent is implemented with a deep neural net. Experiments are performed with head-to-heads against a replication of ReBeL and against Slumbot 2019. Experiments are also performed with exploitability calculations on river endgames, compared against the replication of ReBeL. RL-CFR beats the ReBeL replication on all experiments.

**Strengths:**

The research direction is interesting: using RL to find action abstractions for EFG solving, especially depth-limited EFG solving. There hasn't been much research lately in action abstractions and this direction is (to my knowledge) a novel one that intuitively seems like a great idea.

The engineering effort involved in the experiments must have been rather large, and time-consuming. The experiments ran are all useful. The head-to-head experiments ran (Table 1) are good and not easy to do. The river endgame exploitability experiments are also good to know. The comparisons in Table 2 are also good.

The empirical results are strong.

The descriptions and pseudocode of the algorithm are not bad and are fairly clearly communicated for the most part.

**Weaknesses:**

Clearly a lot of work has been put into this paper. Replicating ReBeL on NLHE alone is a major endeavor. Much engineering effort must have gone into implementing and training RL-CFR and evaluating it against ReBeL and Slumbot. The idea is fresh and the direction seems good, and the results are good. This paper deserves to be published.

However, as it is I am giving this paper a 5 (marginally below the acceptance threshold). I think the paper needs to be cleaned up a lot: the explanations are sometimes confusing, underspecified, or incorrect. In addition, the lack of a clearly stated objective (much less any theory showing that this work achieves the objective) means that this may not meet the bar for ICLR acceptance.

- The paper does not address any theory or optimality. Surely the resulting policy's relation to a Nash equilibrium is, if not a goal, at least an important question. I'm not saying that there needs to be theory saying that this will converge to a Nash equilibrium, but it should at least be mentioned in the paper. The paper mentions that the method here will lead to "performance improvement" (and obviously it's implicit in this paper that the goal is to get a better policy), but it's not described anywhere what this means or why this method will lead to it.

- In lieu of any **theory** showing that this method should lead to a stronger or more optimal policy, there should be some **intuition** for why this method would lead to a stronger or more optimal policy. However, there is none given in the paper. *In particular, I have no intuition for why the given definition of reward (Section 4, page 6) will lead to a better policy in the game.* (Yes, it's intuitive that one way to increase the root PBS value is by picking an action abstraction that lets the turn player compute a less-exploitable strategy. But another way to increase the root PBS value would be by picking an action abstraction that prevents the other player from computing a less-exploitable strategy. It seems we should only care about the former, not the latter. But might maximizing the latter interfere with maximizing the former?)
  - This method clearly seems to perform better, as per the experiment results. Can you speculate on why RL-CFR seems to do so much better?

- The abstract and introduction state that "RL-CFR defeats ReBeL, one of the best fixed action abstraction-based HUNL algorithms". Unfortunately, I think it would be more accurate to clarify that these head-to-head experiments were performed against a replication of ReBeL, since Table 1 shows that the replication achieves a lower head-to-head winrate against Slumbot than the ReBeL from Brown et al., 2020.

- The motivation for wanting to use RL in the Introduction doesn't really make sense to me. The motivation for using RL seems to be hinting towards wanting to use RL for the sake of using RL, rather than for some well-defined reason. I suppose this calls back to my earlier point: it's not clear what the goal is. Is the goal to create an agent that plays a lower-exploitability strategy (i.e. closer to Nash equilibrium)? Then how does the motivation in the Introduction ("Reinforcement learning has been shown to be a revolutionary method in many games") connect to this goal?
  - Similarly, phrasing elsewhere seems to imply that we have as a presupposed goal the desire to implement Deep RL somehow for IIEFGs. But why do we have this goal in the first place? See: Section 5 "It is important to note that applying the DRL approach to IIEFGs is highly nontrivial. The key challenge comes from the fact that one has to decide the action probability distributions for all information sets..." -- I don't get the point of this sentence (also, the "key challenge" isn't super clear to me).

- Section 4 (State): This section should clarify that the PBS to PS reduction is lossy. Indeed, while I think that defining state as the PBS satisfies the Markov assumption, I'm not so sure if PS as state does. If convergence or optimality were touched upon in this paper (they should be!) then this choice of PS rather than PBS may introduce a problem.

- The paper would benefit greatly from qualitative results showing examples of action abstractions chosen by the RL agent.


Minor:

- Section 3 (Public Belief State): "In general, a PBS is described by the joint probability distribution of the possible infostates of the players", but "PBS $\beta$" seems to be defined as the marginal probability distributions for each player. In poker, you can go from the marginal distributions to the joint distribution, but this is not always the case, right? If so, this should be clarified.

- Section 4 (MDP definition): The state, actions, and rewards are defined for this MDP, but as far as I can see, the state transitions are not defined anywhere. I can read between the lines and infer from Algorithm 1, but it would be much clearer if it were defined here.

Nitpicks / typos:

- Introduction typo: "may depends on" should be "may depend on"
- Introduction: "To tackle the above challenge" -- should it be "To tackle the above challenges"? As-is, it implies that RL-CFR tackles the problem of finding a Nash equilibrium in a game. However, ReBeL alone already solves this problem. With the plural "challenges" it would mean that RL-CFR tackles both the problem of finding the Nash equilibrium in a game, and also the problem of picking an action abstraction. Perhaps it would also be clearer to clarify that ReBeL alone already "handles the aforementioned mixed-strategy and probability-dependent reward issues".
- Introduction: "by significantly win-rate margins" should be "by significant margins" or "by significant win-rates"?
- Section 2: The acronym "EGT" should be expanded.
- Section 2: RL with regularization-based payoff functions should also cite:
   - "A Unified Approach to Reinforcement Learning, Quantal Response Equilibria, and Two-Player Zero-Sum Games", Sokota et al (MMD)
   - Neurd and R-nad and stratego:
       - "From poincaré recurrence to convergence in imperfect information games: Finding equilibrium via regularization", Perolat et al.
       - "Neural Replicator Dynamics: Multiagent Learning via Hedging Policy Gradients", Hennes et al.
       - "Mastering the Game of Stratego with Model-Free Multiagent Reinforcement Learning", Perolat et al.
  - In Section 3, quotation marks around "player" are facing the same direction.
- Section 3 (PBS): "At the beginning of a subgame, a state is sampled..." -- should this be "a history is sampled"? Since "state" is ambiguous (being used in the previous sentence in a different sense), whereas a history is rigorously defined.
- Section 4: "... is used to select an action abstract" should be "... is used to select an action abstraction"
- Section 5: "non-chance and non-terminate" should be "non-chance and non-terminal". Same for Footnote 5.
- Section 5 "transform a high-dimensional PBS $\beta$ into a low-dimensional public state $s$": I think this should be more clear that the "transformation" is a lossy one. Maybe by using a more specific verb than "transform"?
- Section 6: "our conduct experiments" should be "we conduct experiments"
- Section 6: It would be helpful to clarify which CFR variation is used in the experiments here. (I see that it is clarified in the Appendix, but it would be useful to note here as well.)
- Section 6: typo: "We evaluate the performance of RL-CFR and ReBeL under the common knowledge in HUNL."
- Section 6: typo: "the agent know" should be "the agent knows"
- Section 6: "We evaluate the performance of RL-CFR and ReBeL" should clarify that this paragraph is regarding head-to-head evaluations of RL-CFR vs. ReBeL, not merely comparisons between them on some common metric. (That's what this paragraph is about, right?) And "RL-CFR achieves 64 mbb/hand win-rate compared to the replication of ReBeL" should be "versus the replication of ReBeL" instead of "compared to the replication of ReBeL".
- Section 6: typo: "We also comapre RL-CFR" should be "compare"
- Appendix D: "In fact, the origin" should be "original"
- Appendix D: typo: "Brown & Sandholm (2016a)" should be in parentheses?
- Appendix D: The second paragraph describes depth-limited solving, but Libratus is mentioned, even though it doesn't do any depth-limited subgame solving.
- Appendix D: "resolve the strategy based on new action abstraction" -- should this also cite the original ReBeL paper?
- Appendix E: Figure 3: typos "Non-Ternimal", "Ternimal Node", "In each iterator" (should be "iteration")
- Appendix E: typo: Footnote 19 in the main text is displayed as ". 19."

**Questions:**

- I think the river endgame experiments are interesting, and would like more details. Do you have 95% confidence intervals for the results? Can you expand on footnote 9: did you play each scenario twice: once as normal, and once where ReBeL and RL-CFR switch places? Could you do river endgames sampled from ReBeL vs. ReBeL preflop-through-turn, or RL-CFR vs. RL-CFR preflop-through-turn, instead of mixed?

- The abstract and Introduction say that "RL-CFR effectively trades off computational complexity (due to CFR) and performance improvement (due to RL)". What is this trade-off? Does RL-CFR have less computational complexity and more performance improvement? This seems like it might be excessively hand-wavy?

- In the introduction: "RL-CFR has a wider range of applicability and faster convergence". Where is this faster convergence experimentally demonstrated or proved? What convergence is being referred to here?

- In Introduction: RL-CFR achieves "a good balance between computation and optimism". Is this expanded upon in the paper? I would love to hear more about this.

- Section 4: "Our design is inspired by (Brown et al., 2019), which transforms high-dimensional public belief states into low-dimensional public states." Can you expand on this? I couldn't find what you were referring to.

- Do we use a discount factor for the RL agent in experiments? Is it just 1?

- In the original ReBeL paper, they show that they can simplify the PBS from perfect recall (encoding the previous actions of the two players) to imperfect recall (only encoding the stack sizes of the two players). In this paper, footnote 3 implies that the public state used here does encode the previous actions of the two players. Is this true?

- It's said in Section 4 (State) that the PBS dimensionality is very large, so we reduce it to the PS for our MDP. However, in terms of deep RL, is the 2,500 dimensionality really a problem? Would it have caused the experiment time to increase by a lot if the PBS was used instead of PS? Also, the PBS value net which takes in a PBS is still trained during ReBeL replication self-training, and used for inference during RL-CFR, right? So doesn't that mean that a deep neural net which takes a PBS as input is tractable?

- In Section 4 (State): "The selection of public states has the additional advantage that the public states of the non-root nodes are fixed during the CFR iterations..." Why does this matter? If I understand correctly, the PS is only needed for the root node, in order to get the action abstraction *before* CFR is started.

- Will the code be open-sourced? It's difficult to evaluate whether everything was implemented without error because there are so many possible pitfalls with implementing a complex system like ReBeL.

- In fact, the head-to-head results versus Slumbot imply that the ReBeL replication does not match ReBeL from the original paper. Do you know or hypothesize why this is?

- Why do you let the action abstraction agent choose to have fewer than K additional action abstractions? Should it not always be "better" to have more bet sizes in your abstraction?

- Section 6: Paragraph 3 (PBS value network training) -- was this done via self-play as in ReBeL? If so, are the details of the training process exactly the same as ReBeL?
  - As just one example of a detail: are the depths of the subgames during self-play defined exactly as they are in the ReBeL paper?

- Section 6: "In addition, the PBS value networks used for all our experiments are trained based on the default action abstraction." Does this mean that randomly modifying bet sizes during self-play as in the original ReBeL paper was not performed?

- Section 5: (2): adding a Gaussian noise -- was this described earlier in the paper? Why is this done?

- Section 5: "... we can retrain the PBS value network according to the action abstraction selected by the action network." and "Theoretically, the PBS value network and action network can be repeatedly updated for training." Can you clarify whether these two sentences refer to the experiments in this paper, or to potential future experiments?

- Algorithm 1: Are the hyperparameters set to 0 during test-time?

- Algorithm 1 differs from ReBeL in that it samples an action at the root and then repeats (constructing the subgame, solving it, and then sampling an action), whereas ReBeL must play the strategy until the end of the subgame before constructing a new subgame. By re-performing search every iteration, the guaranteed convergence towards a Nash equilibrium strategy is lost. Was this considered when designing RL-CFR? Why not play the computed strategy until the end of the subgame like in ReBeL?

- Algorithm 1: What depth do we solve to?

- Section 6: Can you expand on the common knowledge in HUNL? What does it mean that they know each other's historical actions? As in the previous actions played during the hand? But that's always true. Why would we assume that the agents know each other's hand ranges?
  - "Hence, we can avoid actions that are not in the action abstraction." What does this mean? Whose action abstraction? Why can we avoid them?
  - What does it mean that the agents know each other's hand ranges? Concretely, what does this mean in terms of the ReBeL and RL-CFR algorithms used? Does it mean that the PBS used by ReBeL and RL-CFR are set to be the actual PBS based on the action probabilities of the opponent on the previous action? If so, why do this? If so, how do the three references (Burch et al. 2018, Li et al. 2020, Kovarik and Lisy 2019) support this? If this is simply referring to AIVAT, then wouldn't it be more accurate to say that the *evaluator* knows both player's hand ranges, not that the *agent(s?)* know each other's hand ranges?

- Section 6: "Since the opponent may select actions that deviate from the game tree, we perform nested subgame solving..." I don't see why this is the case in Slumbot vs. RL-CFR and not in ReBeL vs. RL-CFR. In the latter, couldn't ReBeL play some bet size that wasn't in RL-CFR's abstraction? Similarly, RL-CFR could play some bet-size that wasn't in ReBeL's abstraction, no?

- Can you clarify MUL-ACTION and FINE-GRAIN? Why only set the action abstractions for the root? Why not for the whole subtree?

- Appendix E: Algorithm 2: What is AAlimit?

---

> ### Author Response · Authors · 2023-11-20
>
> Thank you very much for the valuable comments! We will answer your questions below.
>
> ***
> * Q1. The paper does not address any theory or optimality. Surely the resulting policy's relation to a Nash equilibrium is, if not a goal, at least an important question. I'm not saying that there needs to be theory saying that this will converge to a Nash equilibrium, but it should at least be mentioned in the paper. The paper mentions that the method here will lead to "performance improvement" (and obviously it's implicit in this paper that the goal is to get a better policy), but it's not described anywhere what this means or why this method will lead to it.
>
> * A1. We agree with the reviewer that this should be clarified. We have modified the paper accordingly. Please see the last sentence of Section 1, Paragraph 2 and the last sentence of conclusion in the revision.
>
> ***
>
> * Q2. In lieu of any theory showing that this method should lead to a stronger or more optimal policy, there should be some intuition for why this method would lead to a stronger or more optimal policy. However, there is none given in the paper. In particular, I have no intuition for why the given definition of reward (Section 4, page 6) will lead to a better policy in the game. (Yes, it's intuitive that one way to increase the root PBS value is by picking an action abstraction that lets the turn player compute a less-exploitable strategy. But another way to increase the root PBS value would be by picking an action abstraction that prevents the other player from computing a less-exploitable strategy. It seems we should only care about the former, not the latter. But might maximizing the latter interfere with maximizing the former?)
>
>   * This method clearly seems to perform better, as per the experiment results. Can you speculate on why RL-CFR seems to do so much better?
>
> * A2. The reason we did not use exploitability as reward is because it is difficult to compute if we cannot enumerate all possible actions. Also, in our work, we let the opponent also use the action abstraction chosen by RL-CFR in the second half of the training, which allows our agent to train an action abstraction that is not easily exploitable. We have included the explanation of why RL-CFR seems to do so much better in Footnote 1 of the revised version.
>
> ***
>
> * Q3. The abstract and introduction state that "RL-CFR defeats ReBeL, one of the best fixed action abstraction-based HUNL algorithms". Unfortunately, I think it would be more accurate to clarify that these head-to-head experiments were performed against a replication of ReBeL, since Table 1 shows that the replication achieves a lower head-to-head winrate against Slumbot than the ReBeL from Brown et al., 2020.
>
> * A3. Thanks for the suggestion. We have made this clear in the revision.
>
> ***
>
> * Q4. The motivation for wanting to use RL in the Introduction doesn't really make sense to me. The motivation for using RL seems to be hinting towards wanting to use RL for the sake of using RL, rather than for some well-defined reason. I suppose this calls back to my earlier point: it's not clear what the goal is. Is the goal to create an agent that plays a lower-exploitability strategy (i.e. closer to Nash equilibrium)? Then how does the motivation in the Introduction ("Reinforcement learning has been shown to be a revolutionary method in many games") connect to this goal?
>
> * A4. The goal of RL-CFR is choosing an action abstraction with highest root PBS value (we emphasized this point in Section 1, Paragraph 4). Based on the characteristics of IIEFG, RL has several benefits over other methods. First, there are many different states in IIEFG, and RL is able to discover ways to select action abstractions among these different states. Secondly, each action abstraction in IIEFG can be calculated as a payoff based on CFR, and the optimisation objective of the RL method is very clear. Thirdly, an IIEFG with common knowledge has Markov state transition properties, which make it feasible to use RL for IIEFGs. These features are very similar to some of the games that RL has already solved so we try to apply RL in IIEFGs. Indeed, the experimental results also show that RL's approach achieves a performance improvement by choosing a better action abstraction.

---

> > ### Author Response · Authors · 2023-11-20
> >
> > ***
> >
> > * Q5. Similarly, phrasing elsewhere seems to imply that we have as a presupposed goal the desire to implement Deep RL somehow for IIEFGs. But why do we have this goal in the first place? See: Section 5 "It is important to note that applying the DRL approach to IIEFGs is highly nontrivial. The key challenge comes from the fact that one has to decide the action probability distributions for all information sets..." -- I don't get the point of this sentence (also, the "key challenge" isn't super clear to me).
> >
> > * A5. In the previous problem we mentioned several properties of IIEFGs that make RL a very suitable solution. Thus, we propose a RL framework for solving IIEFGs. However, IIEFG is very different from those common games. While common games only need to choose an optimal action for the current state, IIEFG needs to compute a mixed strategy for all feasible actions for all information sets. It is difficult to get such a mixed strategy directly by RL, so we propose the method of selecting the action abstraction by RL and computing the strategy by CFR. We have modified this sentence in the revision.
> >
> > ***
> >
> > * Q6. Section 4 (State): This section should clarify that the PBS to PS reduction is lossy. Indeed, while I think that defining state as the PBS satisfies the Markov assumption, I'm not so sure if PS as state does. If convergence or optimality were touched upon in this paper (they should be!) then this choice of PS rather than PBS may introduce a problem.
> >
> > * A6. The compression process from PBS to PS results in a loss of information. However, we believe that such compression is necessary and has no impact on RL performance. Since the PS contains the previous actions of both players, if both players perform an equilibrium strategy, we can infer the PBS based on these actions. PBS is necessary for CFR, but not for the action network used to select an action abstraction. Because when using PS, we are only choosing the action abstraction without CFR solving. Thus,  we believe  it is reasonable to use PS as the state in the MDP. Later in the rebuttal we will also mention the problems that can result from choosing PBS as the state in the MDP.
> >
> > ***
> >
> > * Q7. The paper would benefit greatly from qualitative results showing examples of action abstractions chosen by the RL agent.
> >
> > * A7. We will revised it in appendix in final revision.
> >
> > ***
> >
> > * Q8. Section 3 (Public Belief State): "In general, a PBS is described by the joint probability distribution of the possible infostates of the players", but "PBS" seems to be defined as the marginal probability distributions for each player. In poker, you can go from the marginal distributions to the joint distribution, but this is not always the case, right? If so, this should be clarified.
> >
> > * A8. Thank you very much for your help. We have fixed this sentence in revision.
> >
> > ***
> >
> > * Q9. Section 4 (MDP definition): The state, actions, and rewards are defined for this MDP, but as far as I can see, the state transitions are not defined anywhere. I can read between the lines and infer from Algorithm 1, but it would be much clearer if it were defined here.
> >
> > * A9. We have made this clear in the last paragraph of Section 4 of the revision.
> >
> > ***
> >
> > * Q10. Minor/Nitpicks/typos:
> >
> > * A10. Thank you very much for your help. We have fixed these typos.
> >
> > ***
> >
> > * Q11. I think the river endgame experiments are interesting, and would like more details. Do you have 95\% confidence intervals for the results? Can you expand on footnote 9: did you play each scenario twice: once as normal, and once where ReBeL and RL-CFR switch places? Could you do river endgames sampled from ReBeL vs. ReBeL preflop-through-turn, or RL-CFR vs. RL-CFR preflop-through-turn, instead of mixed?
> >
> > * A11. We played each scenario twice and switch places. The detailed results are as follows, with RL-CFR exploiting ReBeL by 20$\pm$0 mbb/hand, ReBeL exploiting RL-CFR by 17$\pm$0 mbb/hand, ReBeL exploiting ReBeL by 16$\pm$0 mbb/hand and RL-CFR exploiting RL-CFR by 16$\pm$0 mbb/hand.

---

> > > ### Author Response · Authors · 2023-11-20
> > >
> > > ***
> > > * Q12. The abstract and Introduction say that "RL-CFR effectively trades off computational complexity (due to CFR) and performance improvement (due to RL)". What is this trade-off? Does RL-CFR have less computational complexity and more performance improvement? This seems like it might be excessively hand-wavy?
> > >
> > > * A12. RL-CFR has a similar computational complexity with ReBeL, and outperforms ReBeL with the same solving time. This is because RL-CFR selects an action abstraction that has a higher expected value calculated by CFR compared to the fixed action abstraction, while at the same time the size of this selected action abstraction does not exceed that of the fixed action abstraction. We have included the explanation of "trade-off" in footnote 1 of the revised version.
> > > ***
> > > * Q13. In the introduction: "RL-CFR has a wider range of applicability and faster convergence". Where is this faster convergence experimentally demonstrated or proved? What convergence is being referred to here?
> > >
> > > * A13. Faster convergence here means that compared to these previous methods of choosing dynamic action abstractions, RL-CFR converges faster because it selects an action abstraction that is fixed during CFR solving. In page 102 of [1], it has been demonstrated "the algorithm converges more slowly than running CFR with fixed bet sizes". Convergence there refers to the exploitability at the same number of iterations in the CFR solving. The smaller the exploitability, the faster the convergence.
> > > ***
> > > * Q14. In Introduction: RL-CFR achieves "a good balance between computation and optimism". Is this expanded upon in the paper? I would love to hear more about this.
> > >
> > > * A14. RL-CFR can achieve performance improvement without increasing the solving time. This is also illustrated by the experimental results in Table 2.
> > > ***
> > > * Q15. Section 4: "Our design is inspired by (Brown et al., 2019), which transforms high-dimensional public belief states into low-dimensional public states." Can you expand on this?
> > >
> > > * A15. For example, in HUNL, the state of PS includes public cards information, chips information, player position information, and previous actions of this hand. A similar state definition for HUNL has also be used in (Brown et al., 2019). We have expanded on this in Footnote 4 of the revised version.
> > > ***
> > > * Q16. Do we use a discount factor for the RL agent in experiments? Is it just 1?
> > >
> > > * A16. Yes. We use a discount factor value 1 in our experiments.
> > > ***
> > > * Q17. In the original ReBeL paper, they show that they can simplify the PBS from perfect recall (encoding the previous actions of the two players) to imperfect recall (only encoding the stack sizes of the two players). In this paper, footnote 3 implies that the public state used here does encode the previous actions of the two players. Is this true?
> > >
> > > * A17. As the origin ReBeL paper mentioned, we can simplify the PBS from perfect recall to imperfect recall, and in our experiments our PBS do not includes previous actions of the two players as well. However, the public state (PS) is different from PBS, and a public state includes the public information that each player knows, and the public information includes the previous actions of the two players.
> > > ***
> > > * Q18. It's said in Section 4 (State) that the PBS dimensionality is very large, so we reduce it to the PS for our MDP. However, in terms of deep RL, is the 2,500 dimensionality really a problem? Would it have caused the experiment time to increase by a lot if the PBS was used instead of PS? Also, the PBS value net which takes in a PBS is still trained during ReBeL replication self-training, and used for inference during RL-CFR, right? So doesn't that mean that a deep neural net which takes a PBS as input is tractable?
> > >
> > > * A18. It would caused the experiment time to increase by a lot if the PBS was used instead of PS. We hypothesize that this reason is the difference in sampling efficiency between ReBeL training and RL-CFR training. In ReBeL a PBS data contains the values of all infostates (2652 dimensions for HUNL), however in RL-CFR a RL data contains only a value scalar about the action abstraction.

---

> > > > ### Author Response · Authors · 2023-11-20
> > > >
> > > > ***
> > > > * Q19. In Section 4 (State): "The selection of public states has the additional advantage that the public states of the non-root nodes are fixed during the CFR iterations..." Why does this matter? If I understand correctly, the PS i s only needed for the root node, in order to get the action abstraction before CFR is started.
> > > >
> > > > * A19. We can set action abstraction by RL-CFR for non-root nodes by PS. In the second half of the training, we let the action abstractions of the root's son nodes to be selected by RL-CFR. Since the PBS changes in each iteration during CFR solving, if we set the PBS to MDP's state, the action abstraction selected by RL-CFR for a non-root node could be changed in each iteration, which will lead to a slower convergence.
> > > > ***
> > > > * Q20. Will the code be open-sourced? It's difficult to evaluate whether everything was implemented without error because there are so many possible pitfalls with implementing a complex system like ReBeL.
> > > >
> > > > * A20. We will open-sourcing the code after the paper is published.
> > > > ***
> > > > * Q21. In fact, the head-to-head results versus Slumbot imply that the ReBeL replication does not match ReBeL from the original paper. Do you know or hypothesize why this is?
> > > >
> > > > * A21. The main reason is that the solving time and training time of ReBeL's replication is smaller compared to the original version. Firstly, the number of actions in our chosen action abstraction may differ from that in ReBeL: we choose up to 6 actions and ReBeL up to 8 actions. Secondly, the replication's iteration is 250, while ReBeL's iteration is 1,024. Thirdly, the size of replication's training data is 48,000,000 while the size of ReBeL's training data is about 750,000,000. We plan to train the replication for more time and see whether the performance improves. Finally, despite the fact that ReBeL's replication did not perform as well as ReBeL, our RL-CFR algorithm, which is based on the ReBeL's replication, achieves a better performance than ReBeL against Slumbot.
> > > > ***
> > > > * Q22. Why do you let the action abstraction agent choose to have fewer than K additional action abstractions? Should it not always be "better" to have more bet sizes in your abstraction?
> > > >
> > > > * A22. An increase in the number of actions in the action abstraction leads to a slower convergence, and a decrease in the number of actions leads to a larger distance to the equilibrium strategy. We choose fewer than K additional action abstractions to obtain a good balance between the speed of convergence and the quality of the solution. It is not always be "better" to have more bet sizes in my abstraction. For example, if small blind player raises and big blind player calls in pre-flop stage. For most public cards in flop stage, the optimal action abstraction for the big blind player should consist of only one action, i.e. check.
> > > > ***
> > > > * Q23. Section 6: Paragraph 3 (PBS value network training) -- was this done via self-play as in ReBeL? If so, are the details of the training process exactly the same as ReBeL?
> > > >   * As just one example of a detail: are the depths of the subgames during self-play defined exactly as they are in the ReBeL paper?
> > > >
> > > > * A23. It was done via self-play as in ReBeL, and the depths of the subgames during self-play defined exactly as they are in the ReBeL paper. The training settings are slightly different from those in the RebeL paper, such as the number of sampling epochs, the number of samples sampled per epoch, and the default action abstraction.
> > > > ***
> > > > * Q24. Section 6: "In addition, the PBS value networks used for all our experiments are trained based on the default action abstraction." Does this mean that randomly modifying bet sizes during self-play as in the original ReBeL paper was not performed?
> > > >
> > > > * A24. We multiply the chips in the pot by a random number between 0.9 and 1.1 during training the PBS value network. This is explained in Footnote 22.
> > > > ***
> > > > * Q25. Section 5: (2): adding a Gaussian noise -- was this described earlier in the paper? Why is this done?
> > > >
> > > > * A25. This is to increase the exploration of the action space. We have explained it in the revised version based on your comments.

---

> > > > > ### Author Response · Authors · 2023-11-20
> > > > >
> > > > > ***
> > > > > * Q26. Section 5: "... we can retrain the PBS value network according to the action abstraction selected by the action network." and "Theoretically, the PBS value network and action network can be repeatedly updated for training." Can you clarify whether these two sentences refer to the experiments in this paper, or to potential future experiments?
> > > > >
> > > > > * A26. These two sentences refer to potential future experiments. We explain clearly the specific experimental setting in the experiment section.
> > > > > ***
> > > > > * Q27. Algorithm 1: Are the hyperparameters set to 0 during test-time?
> > > > >
> > > > > * A27. Yes, these hyperparameters are set to 0 during test-time.
> > > > > ***
> > > > > * Q28. Algorithm 1 differs from ReBeL in that it samples an action at the root and then repeats (constructing the subgame, solving it, and then sampling an action), whereas ReBeL must play the strategy until the end of the subgame before constructing a new subgame. By re-performing search every iteration, the guaranteed convergence towards a Nash equilibrium strategy is lost. Was this considered when designing RL-CFR? Why not play the computed strategy until the end of the subgame like in ReBeL?
> > > > >
> > > > > * A28. This is due to the different objectives of RL-CFR and ReBeL. Specifically, our goal is to give a reasonable action abstraction for any state in the IIEFG, whereas the goal of ReBeL is to give PBS value estimates for a number of specified phases of the IIEFG. Since we need to choose action abstraction for all states, we have to sample an action at root and then repeats.
> > > > > ***
> > > > > * Q29. Algorithm 1: What depth do we solve to?
> > > > >
> > > > > * A29. In Footnote 18, we build the subgame up to the end of the two players' actions in a stage or the end of the chance player's action.
> > > > > ***
> > > > > * Q30. Section 6: Can you expand on the common knowledge in HUNL? What does it mean that they know each other's historical actions? As in the previous actions played during the hand? But that's always true. Why would we assume that the agents know each other's hand ranges?
> > > > >
> > > > > * A30. We assume the agents know the previous actions played during this hand, and that the agents know each other's hand ranges to make the calculation of PBS easier. This does not affect the accuracy of the experiment, because what we do is simply transmit the same PBS to two agents. We have clarified this sentence in the revision.
> > > > >
> > > > > * Q31. "Hence, we can avoid actions that are not in the action abstraction." What does this mean? Whose action abstraction? Why can we avoid them?
> > > > >
> > > > > * A31. This is to show that we can avoid using nested subgame solving techniques (avoid actions not in the action abstraction of acting player), because the acting player needs to declare his mixed strategy publicly. We have revised the sentences and Footnote 11 in revision.
> > > > >
> > > > > * Q32. What does it mean that the agents know each other's hand ranges? Concretely, what does this mean in terms of the ReBeL and RL-CFR algorithms used? Does it mean that the PBS used by ReBeL and RL-CFR are set to be the actual PBS based on the action probabilities of the opponent on the previous action? If so, why do this? If so, how do the three references (Burch et al. 2018, Li et al. 2020, Kovarik and Lisy 2019) support this? If this is simply referring to AIVAT, then wouldn't it be more accurate to say that the evaluator knows both player's hand ranges, not that the agent(s?) know each other's hand ranges?
> > > > >
> > > > > * A32. During evaluation between ReBeL and RL-CFR, the acting player needs to declare his mixed strategy publicly, so that the agents know each other's hand ranges. It means that PBS is public information and the PBS used by ReBeL and RL-CFR are set to be the actual PBS updated by mixed strategy and actual actions. This setting is mainly used to avoid using nested subgame solving techniques. We have revised this sentence in Footnote 11 in revision.

---

> > > > > > ### Author Response · Authors · 2023-11-20
> > > > > >
> > > > > > ***
> > > > > > * Q33. Section 6: "Since the opponent may select actions that deviate from the game tree, we perform nested subgame solving..." I don't see why this is the case in Slumbot vs. RL-CFR and not in ReBeL vs. RL-CFR. In the latter, couldn't ReBeL play some bet size that wasn't in RL-CFR's abstraction? Similarly, RL-CFR could play some bet-size that wasn't in ReBeL's abstraction, no?
> > > > > >
> > > > > > * A33. It is worth emphasising that the evaluation experiment is fair for both agents. In particular, ReBeL and RL-CFR can both play bet sizes not in each others' abstraction. But when the agents act, they assign their hand range to the chosen action abstraction, so the PBS is mutually known to both agents.
> > > > > > ***
> > > > > > * Q34. Can you clarify MUL-ACTION and FINE-GRAIN? Why only set the action abstractions for the root? Why not for the whole subtree?
> > > > > >
> > > > > > * A34. MUL-ACTION is to choose an action abstraction with the maximum expected CFV among 3 action abstractions, and FINE-GRAIN is a larger action abstraction. Our goal is to choose a reasonable action abstraction for the current action player, and the root node corresponds to the current action player, so the variable we experiment with is the root node's action abstraction. In addition, to choose more action abstractions for the whole game tree would increasing the solving time.
> > > > > > ***
> > > > > > * Q35. Appendix E: Algorithm 2: What is AAlimit?
> > > > > >
> > > > > > * A35. Sorry it is a typo. It should be  AA. We have revised it.
> > > > > > ***
> > > > > >         [1] Noam Brown. Equilibrium Finding for Large Adversarial Imperfect-Information Games. PhD thesis, Carnegie Mellon University, 2020.
> > > > > >
> > > > > > We hope our responses address your concerns. Should you find the rebuttal satisfying, we wonder if you could kindly consider raising the score rating for our paper? We will also be happy to answer any further questions you may have. Thank you very much.

---

> > > > > > > ### Author Response · Authors · 2023-11-21
> > > > > > >
> > > > > > > * A7. The paper would benefit greatly from qualitative results showing examples of action abstractions chosen by the RL agent.
> > > > > > >
> > > > > > > * Q7. Following the reviewer's suggestion, in Appendix G of the revision, we describe several detailed examples of how RL-CFR selects action abstractions.
> > > > > > >
> > > > > > > ***
> > > > > > >
> > > > > > > We wonder whether our responses address your concerns? Should you find the rebuttal satisfying, we wonder if you could kindly consider raising the score rating for our paper? We will also be happy to answer any further questions you may have. Thank you very much.

---

> > > > > > > > ### Comment · Reviewer_3LiM · 2023-11-23
> > > > > > > >
> > > > > > > > Thank you for the very thorough response to my initial review. In particular, the new appendix section G "EXAMPLES OF RL-CFR STRATEGIES" is a very very good addition to the paper.
> > > > > > > >
> > > > > > > > I would like to reemphasize what I said in my initial review: I appreciate the very large amount of time and effort that must have gone into this work. I am not super familiar with action abstraction literature, but it does seem to me that there is still more research to be done in the area and so ideas like the one in this paper are welcome contributions.
> > > > > > > >
> > > > > > > > However, I am still hesitant to recommend this paper in its current state for publication at ICLR because there are still unaddressed issues in its writing/explanations, and because my Q2 (theory or intuition for why this method would perform well) is not satisfactorily addressed.
> > > > > > > >
> > > > > > > > I am confident that a future revision of this paper is something I would recommend for publication at some venue, as the work is valuable. Indeed, with the promise to open-source the code, this work is probably already more valuable than some ICLR papers.
> > > > > > > >
> > > > > > > > Q2: I don't think the answer really addressed my question. Perhaps this is because I misunderstand the work. It's not clear how exactly the chosen action abstraction leads to the subgame (constructsubgame in pseudocode). I assume that if the action abstraction is 0.6x 1x 1.6x and the depth is 3, then all of player 1's raises and player 2's raises in the subgame will use these sizes. Then, wouldn't the chosen definition of reward sometimes choose a very "weak" action abstraction so that player 2 must play poorly? Perhaps another way to frame this is: why not just make the action abstraction apply to player 1 in the subgame but not player 2? I still have no intuition or theory for why this choice of reward is good.
> > > > > > > >
> > > > > > > > Pseudocode: describe constructsubgame
> > > > > > > >
> > > > > > > >
> > > > > > > > The definition of pbs as marginal is not correct. (Plus, I clicked on one of the references, oliehoek, which defines it as joint). My previous comment meant that PBS is indeed joint, but you have incorrectly given the math definition as marginal. These are equivalent in poker, but not in general. That is the clarification that needs to be made.
> > > > > > > >
> > > > > > > > Some new additions need proofreading. For example, new footnote 4, which describes how PBSs in poker need not contain action histories, is not a clear explanation for an unfamiliar reader. It should probably cite and/or repeat the explanation from the ReBeL paper which explains why this is allowed.

---

> ### Author Response · Authors · 2023-11-23
>
> Thank you for your reply and for recognising our work!
>
> ***
>
> Regarding Q2, we would like to clarify that the action abstraction explored during training is only done at the root history, and the action abstractions of the other histories are not determined by this explored action abstraction. Thus, the chosen action abstraction does not make the opponent's strategy weaker.
>
> For example, if the explored action abstraction is 0.6x, 1x, 1.6x and the depth is 3, the root's action abstraction is 0.6x, 1x, 1.6x, while other histories' action abstraction is not 0.6x, 1x, 1.6x. In the first half of RL-CFR's training, other histories will choose an action abstraction 0.5x, 1x, 2x. In the second half of the training, other histories will then use the action abstraction chosen by RL-CFR without adding Gaussian noise.
>
> To avoid confusion, we have clarified it in Footnote 6 in the new revision.
>
> ***
>
> As for other questions, we have described constructsubgame in Algorithm 2 in the new revision. We have fixed what was incorrect about the definition of PBS and expanded the explanation in Footnote 4 in new revision.
>
> Please let us know if you have any further questions. We will be happy to answer them. If you find our responses satisfying, could you kindly consider raising the score rating for our work?
>
> Thank you very much!

---

### Meta-Review · Area_Chair_xwyq · 2023-12-06

**Metareview:**

This was a highly borderline paper with high-quality reviews and a robust engagement.

Overall, while I think with revision this paper would be rather strong, I'm leaning towards rejection in the current state. One reservation is about the presentation, even despite the detailed reviews:
- In light of the lack of guarantees (Discussion with reviewer 3LiM), perhaps the adjective "Optimal" in the title could be revisited
- "Public belief state (PBS) is an assumption that treats players’ strategies as common knowledge for reducing the state of large IIEFGs significantly". This is unclear/wrong as stated.
- "Observation-action history is a kind of information set introduced in (Burch et al., 2014)". This is also wrong as stated
- "Our design is inspired by (Brown et al., 2019), which transforms high-dimensional public belief states into low-dimensional public states.". Deep CFR has nothing to do, as far as I know, with public belief states
- I might have missed it, but I was not able to find the mentioned figure of 50mbb/hand in the provided citation here: "Note that a win-rate of over 50 mbb/hand in poker is called a significant win-rate (Bowling et al., 2017),"
- Citing a 2023 paper using category theory for extensive-form games seems really odd. Overall, the choice of citations should be revisited carefully.
- Another example is the choice of citing NFSP among CFR-based algorithms. I do not believethat NFSP is a member of the CFR family.
- Another example is citing Billings et al. among the imperfect-information search literature.

I strongly encourage the authors to take the time to carefully disentangle the citations and make sure they are used carefully.

Finally, another important point that has not been brought up in the discussion is about the computational resources. My understanding is that 4 GPUs are nowhere close to enough to replicate ReBeL results. I believe the authors should comment on this and why this does not throw into discussion the whole empirical conclusions.

**Justification For Why Not Higher Score:**

I think the current version has some important issues that need to be addressed (see meta review)

**Justification For Why Not Lower Score:**

N/A

---

### Decision · Program_Chairs · 2024-01-16

Reject